# RSQ: Learning from Important Tokens Leads to Better Quantized LLMs

**Yi-Lin Sung[1]**                                  *ylsung@cs.unc.edu*
**Prateek Yadav[1]**                                 *praty@cs.unc.edu*
**Jialu Li[1]**                                      *jialuli@cs.unc.edu*
**Jaehong Yoon[2]**                                  *jaehong.yoon@ntu.edu.sg*
**Mohit Bansal[1]**                                  *mbansal@cs.unc.edu*
[1] *UNC at Chapel Hill*
[2] *NTU Singapore*

**Reviewed on OpenReview:** *https://openreview.net/forum?id=kBezrKXHVS*

## Abstract

Layer-wise quantization is a key technique for efficiently compressing large models without expensive retraining. Previous methods typically quantize the weights of each layer by "uniformly" optimizing the layer reconstruction loss across all output tokens. However, in this paper, we demonstrate that better quantized models can be obtained by prioritizing learning from important tokens. Building on this finding, we propose RSQ (Rotate, Scale, then Quantize), which (1) applies rotations (orthogonal transformation) to the model to mitigate weight outliers, (2) scales the token feature based on its importance, and (3) quantizes the model using the GPTQ framework with the second-order statistics computed by scaled tokens. To compute token importance, we explore both heuristic and dynamic strategies. Based on a thorough analysis of all approaches, we adopt attention concentration, which uses attention scores of each token as its importance, as the best approach. We demonstrate that RSQ consistently outperforms baseline methods across multiple downstream tasks and three model families: LLaMA3, Mistral, and Qwen2.5. Additionally, models quantized with RSQ achieve superior performance on long-context tasks, further highlighting its effectiveness. Lastly, RSQ demonstrates generalizability across various setups, including different model sizes, calibration datasets, bit precisions, and quantization methods. Our code is available at https://github.com/ylsung/rsq.

## 1 Introduction

Large language models (LLMs) (Georgiev et al., 2024; Achiam et al., 2023) have transformed the landscape of artificial intelligence, but their high computational demands make deployment challenging, especially in resource-constrained settings. Weight quantization (Han et al., 2016; Wu et al., 2016) addresses this by representing weights with fewer bits to reduce compute costs. Among various methods, post-training quantization (PTQ) (Frantar & Alistarh, 2022; Liu et al., 2021) is especially popular, as it quantizes pre-trained LLMs using a small calibration set without expensive retraining.

We focus on layer-wise post-training quantization (Hubara et al., 2020; Li et al., 2021; Frantar et al., 2023), which has proven effective and efficient for large models. These methods quantize weights one layer at a time by minimizing the token-level feature distance between outputs of original and quantized weights (i.e., the reconstruction loss, $\|\mathbf{WX} - \tilde{\mathbf{W}}\mathbf{X}\|_2^2$). Recent improvements include GPTQ (Frantar et al., 2023), which enhances efficiency and stability in computing second-order statistics, and methods like QuIP#(Tseng et al., 2024), AQLM(Egiazarian et al., 2024), and QTIP (Tseng et al., 2025), which use vectors instead of fixed scalars to represent weights. Additionally, QuIP (Chee et al., 2023) and QuaRot (Ashkboos et al., 2024b) demonstrate through empirical studies that weight outliers (parameters with unusually large magnitudes) can be effectively mitigated by applying orthogonal transformations.

Prior methods typically perform layer-wise quantization by uniformly optimizing the reconstruction loss across all input tokens. However, studies show that LLMs do not treat all tokens equally: (1) StreamingLLM (Xiao et al., 2024) finds that initial tokens often have strong attention scores, (2) $H_2O$ (Zhang et al., 2023) reveals that some tokens in KV cache contribute most of the attention values while decoding, and (3) RHO-1 (Lin et al., 2024c) demonstrates not all tokens are equal in training LLMs. Since models inevitably lose information after quantization, we argue that preserving the most critical information is essential for maintaining performance. Inspired by these insights, we reconsider the conventional approach in quantization methods by optimizing the layer reconstruction loss over only a subset of important input tokens (*i.e.*, using only the first 1/4 of the tokens). Our pilot experiments in Sec. 4.1 reveal that this strategy improves the quantized model's accuracy across ten downstream tasks by up to 2.2%.

Building on our findings and previous approaches, we propose RSQ to quantize the model in three steps: (1) *rotate* (orthogonally transform) the model to mitigate weight outliers, (2) *scale* the token feature based on its importance, and (3) *quantize* the weights using GPTQ mechanism while leveraging token importance. We note that token importance integrates seamlessly into several layer-wise quantization frameworks (*e.g.*, GPTQ, QuIP#, and QTIP) during the third step, allowing RSQ to leverage their existing kernels for fast inference. Fig. 1 illustrates the three steps in RSQ.

In this paper, we explore two categories of approaches for determining token importance: (1) heuristic and (2) dynamic approaches. In the heuristic category, we study methods like First-N and First&Last-N, which prioritize initial tokens or a combination of initial and final tokens. These approaches outperform the conventional quantization of optimizing across all tokens, achieving peak performance when N is roughly 5–10% of the total tokens. Since heuristic approaches rely only on position-based rules, we further investigate dynamic approaches that compute token importance per input. Specifically, we investigate ActNorm, which prioritizes tokens with *larger norms*; TokenSim, which gives greater weight to tokens that are *less similar* to others; and AttnCon, where tokens receiving *higher attention* scores are considered more important. Among these, AttnCon performs the best as it more explicitly models each token's impact on the other tokens. Therefore, we adopt it as our final strategy. In these approaches, we observe positional biases, where certain positions consistently receive higher importance. To avoid underutilizing tokens in "less important" positions, we apply a data augmentation strategy that shifts inputs forward by several positions to improve performance.

We evaluate RSQ on a diverse set of tasks using three models: LLaMA3-3B-Instruct, Mistral-Nemo-12B, and Qwen2.5-7B. Our results demonstrate that RSQ achieves absolute improvements of 1.6%, 0.9%, and 0.4% in average accuracy across the tasks for the three models over QuaRot, respectively. Furthermore, we evaluate RSQ on subsets of various long-context benchmarks, including LITM, L-Eval, and LongICLBench. Our results show that RSQ achieves improvements of 2.1%, 2.5%, and 0.6%, respectively, over QuaRot on these benchmarks. Lastly, we demonstrate the generalizability of RSQ across various setups, such as different model sizes, calibration datasets, bit precisions, and quantization methods (including joint quantization and vector quantization). Notably, the performance improvement is more pronounced at lower bit precisions, suggesting that "learning from important token" can be a critical component in pursuing effective extreme compression.

## 2 Related Work

PTQ has gained significant attention for its ability to quantize pre-trained models without requiring expensive retraining. Layer-wise quantization is a core approach for PTQ, with great advancements made in recent years. Specifically, GPTQ (Frantar et al., 2023) and OBC (Frantar & Alistarh, 2022) quantize weights and adjust the remaining weights accordingly with data-derived Hessian matrices (second-order statistics). AWQ (Lin et al., 2024b) minimizes quantization error by rescaling weights based on the activation distribution. QuIP# (Tseng et al., 2024) AQLM (Egiazarian et al., 2024), QTIP (Tseng et al., 2025) represent groups of quantized weights with vectors instead of scalar values. We focus on weight quantization, as model size remains a primary challenge in utilizing large LLMs. One challenge in weight quantization is the presence of outliers in the weights, as LLMs often contain a subset of weights with exceptionally large magnitudes (Dettmers et al., 2024; Kim et al., 2024; Yu et al., 2024). During quantization, most weights are forced into a narrow range to fit these outliers within the quantized representation, which ultimately leads to suboptimal performance. One

way to address this challenge is through mixed-precision quantization (Dettmers et al., 2024; 2022; Kim et al., 2024); however, current hardware lacks efficient support for inference using this technique. Recent studies, such as QuIP (Chee et al., 2023) and QuaRot (Ashkboos et al., 2024b), have empirically shown that outliers in weights can be effectively mitigated by applying orthogonal transformations (rotations) (Ashkboos et al., 2024a), which reduce outliers' impact and minimize the need for mixed-precision quantization. Different from previous advancements, RSQ focuses on improving the learning objective of layer-wise quantization methods (detailed in Sec. 4.2) by prioritizing important tokens. Our approach also builds on QuaRot by applying rotations to mitigate weight outliers and leveraging GPTQ for quantization. This integration makes RSQ a comprehensive and holistic quantization strategy. An expanded discussion of related work is provided in the Appendix.

## 3 Background

### 3.1 Layer-wise Quantization

Layer-wise quantization offers a more efficient alternative to full-model quantization by processing each layer individually. In this approach, the quantized weights for each layer are optimized by minimizing the layer-wise reconstruction loss, which measures the feature distance between the outputs produced by the original weight matrix $\mathbf{W} \in \mathbb{R}^{dout \times din}$ and the quantized weight matrix $\tilde{\mathbf{W}}$: $\|\mathbf{W}\mathbf{X} - \tilde{\mathbf{W}}\mathbf{X}\|_2^2 = \sum_{i=0}^{T} \|\mathbf{W}\mathbf{X}_{:,i} - \tilde{\mathbf{W}}\mathbf{X}_{:,i}\|_2^2$, where $\mathbf{X} \in \mathbb{R}^{din \times T}$ is input tokens' features and $T$ is the sequence length. Note that the reconstruction loss for each token is weighed **uniformly** across a sequence of token features $\{\mathbf{X}_{:,1}, \mathbf{X}_{:,2}, ..., \mathbf{X}_{:,T}\}$. The OBC framework (Frantar & Alistarh, 2022) provides an explicit formula to optimally quantize a column of weight based on the layer-reconstruction loss, as well as the optimal update of the remaining weights which compensates for the quantization. Specifically, for the $q$-th column of weight $\mathbf{W}_{:,q}$ to be quantized, OBC first quantizes the weight in the round-to-nearest manner ($\tilde{\mathbf{W}}_{:,q} = \text{quant}(\mathbf{W}_{:,q})$), and adjust the remaining weights according to the following formula:

$$\boldsymbol{\delta} = -\frac{\mathbf{W}_{:,q} - \text{quant}(\mathbf{W}_{:,q})}{\mathbf{H}_{qq}^{-1}} \cdot \mathbf{H}_{q,:}^{-1} \tag{1}$$

where $\mathbf{H} = 2\mathbf{X}\mathbf{X}^\top$ is the Hessian matrix. After quantizing each layer, we compute its output using the quantized weights, which are then used as input to the next layer.

### 3.2 Removing Outliers with Rotation

Transformer architecture exhibits computational invariance (Ashkboos et al., 2024a), allowing an orthogonal transformation (*a.k.a.* rotation) to be applied to one layer and its transpose to be applied to the subsequent layer without altering the outputs. Specifically, given the input $\mathbf{X}$ and a two-layer module with its output defined as $\mathbf{Y} = \mathbf{W}_2\mathbf{W}_1\mathbf{X}$, it follows that:

$$\mathbf{Y} = (\mathbf{W}_2\mathbf{Q}^\top)(\mathbf{Q}\mathbf{W}_1)\mathbf{X} \tag{2}$$

where both weight matrices, $\mathbf{W}_1$ and $\mathbf{W}_2$ are orthogonally transformed by orthogonal matrix $\mathbf{Q}$ ($\mathbf{Q}^\top\mathbf{Q} = \mathbf{Q}\mathbf{Q}^\top = \mathbf{I}$ by definition). This property remains valid even when an RMSNorm (Zhang & Sennrich, 2019) layer is placed between $\mathbf{W}_1$ and $\mathbf{W}_2$, because $\text{RMSNorm}(\mathbf{Q}\mathbf{W}_1\mathbf{X}) = \mathbf{Q} \cdot \text{RMSNorm}(\mathbf{W}_1\mathbf{X})$. Note that the intermediate features produced by $\mathbf{Q}\mathbf{W}_1$ are also transformed, and their outliers are mitigated similarly.

Prior studies have empirically shown that applying orthogonal transformations to modern pre-trained LLMs can effectively reduce weight outliers (Chee et al., 2023; Ashkboos et al., 2024b). Moreover, due to the computational invariance property, these transformations do not alter the model's output if they are properly inserted into the model. Concretely, assume we initialize an orthogonal transformation matrix $\mathbf{Q}$, which can be a random orthogonal matrix or a randomized Hadamard matrix. We transform the following weight matrices from $\mathbf{W}$ to $\mathbf{W}\mathbf{Q}^\top$: $\mathbf{W}_q$, $\mathbf{W}_k$, $\mathbf{W}_v$ in attention layers, $\mathbf{W}_{up}$, $\mathbf{W}_{gate}$ in FFN layers, and the lm_head layer. Similarly, we transform the following weight matrices from $\mathbf{W}$ to $\mathbf{Q}\mathbf{W}$: $\mathbf{W}_o$ in attention layers, $\mathbf{W}_{down}$

in FFN layers, and the embedding layer. For more details on the rotation and orthogonal transformation, please refer to SliceGPT (Ashkboos et al., 2024a).

## 4 Methodology

### 4.1 Observation: Less Tokens yet Better Performance

As described in the Sec. 3.1, layer-wise quantization methods, such as GPTQ and QuaRot, uniformly minimize the reconstruction loss across all tokens. In the standard setup, using 256 data points from WikiText-2 (Merity et al., 2016), each containing 4096 tokens, to quantize LLaMA3-8B-Instruct (Dubey et al., 2024) to 3-bit precision, QuaRot achieves a perplexity of 9.51 on WikiText-2 and an average accuracy of 63.8% across 10 tasks (detailed in Sec. 5.2).

Here, we present a surprising finding: **using only a subset of tokens can improve performance**. Specifically, we divide the input tokens into four non-overlapping chunks, with each chunk containing 1024 tokens. We then perform four separate quantization runs using QuaRot with applying the reconstruction loss to only one chunk at a time. Notably, while the

Table 1: Quantization results with different subsets.

| Used Token IDs | All: 1 - 4096 | 1 - 1024 | 1025 - 2048 | 2049 - 3072 | 3073 - 4096 |
|---|---|---|---|---|---|
| Wiki PPL ↓ | $9.51_{.11}$ | $\mathbf{9.27}_{.05}$ | $10.26_{.12}$ | $10.16_{.08}$ | $10.25_{.01}$ |
| Avg Acc (%) ↑ | $63.8_{0.5}$ | $\mathbf{64.5}_{0.4}$ | $61.7_{0.7}$ | $61.4_{0.4}$ | $61.3_{0.5}$ |

loss is computed exclusively on the selected chunk, all tokens are used for the forward pass. Therefore, the earlier tokens, even when not selected, still indirectly contribute to the feature of the later tokens. Interestingly, as shown in Tab. 1, the performance when using the 2nd, 3rd, or 4th chunks is inferior to that of the 1st chunk, despite these chunks having access to more input information. We hypothesize that this effect arises because LLMs tend to heavily attend to "the initial tokens", making it crucial to preserve their features by directly minimizing their reconstruction loss. This hypothesis is further supported by broad observations from prior research (Sun et al., 2024a; Xiao et al., 2024). Moreover, we demonstrate that the quantized LLMs **using only the 1st chunk even outperforms the result obtained by using all tokens**. We believe this is because the quantized model has to allocate its limited capacity to learn from all tokens, including those that may be less important than the first chunk.

Based on this observation, we argue that the current approach of uniformly applying reconstruction loss to all tokens is suboptimal, as it fails to prioritize and preserve the most critical information in the model during quantization. To address this, we modify the layer-wise quantization objective to account for token importance and propose RSQ, which we detail in the next section.

### 4.2 RSQ (Rotate, Scale, then Quantize)

In the previous section, we show that a simple approach, selecting only the first chunk, already leads to improved results. Building on this insight, we further explore advanced strategies for assigning importance to different tokens, which we detail in Sec. 4.3. Before delving into these strategies, we first formally introduce the algorithm of RSQ, which quantizes the model in three steps:

**Rotate.** Before quantization, we first mitigate the outliers by rotating the model. As mentioned in Sec. 3.2, computational invariance holds when RMSNorm is used in transformers, such as LLaMA3, Mistral, and Qwen2.5 model families. For models that instead use LayerNorm (Ba et al., 2016), this property does not hold directly. Fortunately, following Ashkboos et al. (2024a), LayerNorm can be converted to RMSNorm by fusing the mean subtraction component to the previous layer and its linear component into the subsequent linear layer. After this conversion, we initialize $\mathbf{Q}$ as a randomized Hadamard matrix (Halko et al., 2011) and apply the rotation matrix to the transformer's weights as described in Sec. 3.2.

**Scale.** As described in Sec. 3.1, previous methods treat the layer reconstruction loss of every token equally, that is $\|\mathbf{WX} - \tilde{\mathbf{W}}\mathbf{X}\|_2^2 = \sum_{i=0}^{T}\|\mathbf{WX}_{:,i} - \tilde{\mathbf{W}}\mathbf{X}_{:,i}\|_2^2$. In contrast, RSQ assigns different importance to different

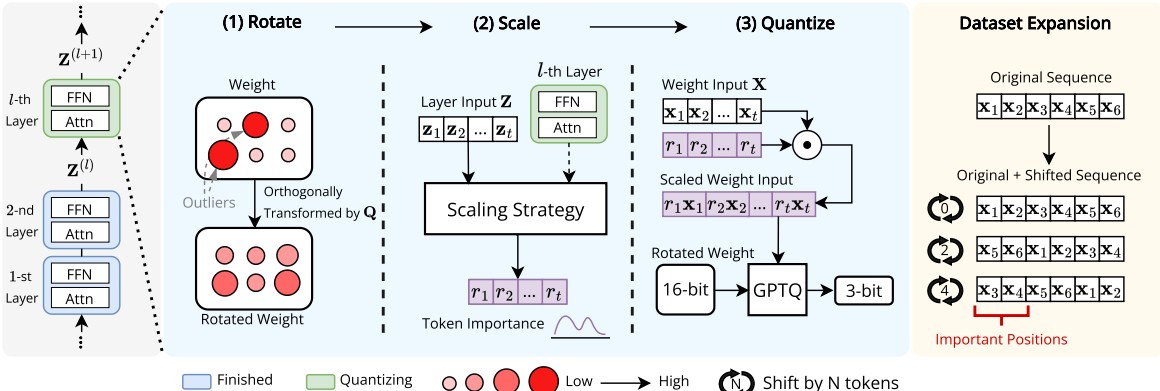

Figure 1: Illustration of layer-wise quantization (left), three-step process of RSQ (middle) and the dataset expansion (right). On the middle, circle size and red color intensity represent weight magnitude, with larger circles and deeper red colors indicating greater magnitudes.

tokens and modifies the objective function accordingly:

$$\|(\mathbf{WX} - \tilde{\mathbf{W}}\mathbf{X})\mathbf{R}\|_2^2 = \sum_{i=0}^{T} \|r_i(\mathbf{WX}_{:,i} - \tilde{\mathbf{W}}\mathbf{X}_{:,i})\|_2^2, \tag{3}$$

where $\mathbf{R}$ is a diagonal matrix with diagonal entries $\{r_1, r_2, ..., r_T\}$ that scale the token representations. The specific methods for assigning values to the importance matrix $\mathbf{R}$ are detailed in the next section.

**Quantize.** Given RSQ's proposed objective function (Eq. (3)), as in Sec. 3.1, we follow the GPTQ framework to solve the optimal quantized weight while minimizing the loss. The resulting formulation remains mostly the same, except for a modified Hessian matrix $\mathbf{H}_{\text{RSQ}} = 2\mathbf{XR}^2\mathbf{X}^\top$, which essentially represents the outer product of the scaled token features (*i.e.* $\mathbf{XR}$). Next, we quantize the rotated weight by applying the modified Hessian matrix to the weight update formula presented in Eq. (1). Fig. 1 displays the illustration of the three steps of RSQ.

## 4.3 Strategies to Compute Token Importance

To align with the nature of layer-wise quantization, we compute token importance per layer independently. Furthermore, we avoid using any global information, such as model gradients, as it would violate the layer-wise assumption, where only one layer is accessed at a time. During the quantization of the $l$-th layer, let $\mathbf{Z}^{(l)} \in \mathbb{R}^{d \times T} = \{\mathbf{Z}_{:,i}^{(l)} : 1 \leq i \leq T\}$ represent the $d$-dimensional input features of the current layer (note that $\mathbf{Z}^{(l+1)} = \texttt{Layer}^{(l)}(\mathbf{Z}^{(l)})$). We compute the token importance matrix $\mathbf{R}^{(l)}$ with diagonal values $\{r_i^{(l)} : 1 \leq i \leq T\}$ ($r \in \mathbb{R}$) to reweight the input feature of the weight in this layer to $\{r_i^{(l)}\mathbf{X}_{:,i}^{(l)} : 1 \leq i \leq T\}$ before applying GPTQ for weight quantization. Note that we use $\mathbf{Z}$ to represent the input features of a "layer", distinguishing it from $\mathbf{X}$, which denotes the input features associated with a "weight." The token importance is kept consistent across all weights within a layer, as we observe this yielding better performance. For clarity and simplicity, we omit the superscript for the layer index $l$ in the remaining of this section.

Next, we present several methods for assigning token importance to complete the second step of RSQ. We start with two *heuristic* approaches that prioritize tokens on specific positions in the sequence.

**First-N.** Building on our observation in Sec. 4.1 that using fewer tokens (specifically, tokens from the first chunk) leads to better performance, we further divide the inputs into smaller chunks, each containing fewer tokens. We then evaluate the quantization performance using only the first chunk. Formally, we define $r_i = 1$ when $i \leq N$, and $r_i = 0$ for the rest.

**First&Last-N.** This approach extends the First-N method. While First-N uses the first and second N/2 tokens, we instead select the first and last N/2 tokens for quantization. We hypothesize that using the last chunk may better capture long-term dependencies. Formally, we assign $r_i = 1$ when $i \leq N$ or $i > T - N$, and $r_i = 0$ otherwise.

The aforementioned approaches assign token importance based on positional heuristics, ignoring variations across different samples and layers. To address this limitation, we explore several *dynamic* approaches where token importance is determined adaptively based on the layer inputs $\mathbf{Z}$ and model characteristics. We rigorously compare these approaches and adopt the most effective one at the end.

Notably, since the score distributions produced by different dynamic approaches can vary significantly, we linearly transform the importance values ($\forall r \in \mathbf{R}$) into a bounded range $[r_{min}, r_{max}]$:

$$r = r_{min} + \frac{r - \min(\mathbf{R})}{\max(\mathbf{R}) - \min(\mathbf{R})} \cdot (r_{max} - r_{min}) \tag{4}$$

Here, $r_{min}$ and $r_{max}$ are hyperparameters, and we always set $r_{max}$ to 1 and adjust $r_{min}$ to vary the emphasis on less important tokens. Note that we apply Eq. (4) to normalize the scores into a bounded range for *all* dynamic approaches. Next, we introduce three of our dynamic strategies in detail (the other two additional strategies are in Appendix).

**Activation Norm (ActNorm).** Previous studies show that inputs with larger norms have a greater impact on the layer's outputs (Virmaux & Scaman, 2018). Sun et al. (2024a) also show that attention in LLMs tends to concentrate on tokens with larger norms. Based on these, we design an approach to assigning importance scores to tokens based on the norm of their input activations, that is $\mathbf{R} = \{\|\mathbf{Z}_{:,i}\| : 1 \leq i \leq T\}$.

**Token Similarity (TokenSim).** This approach assigns token importance based on the pairwise similarity between each token and all other tokens. Our assumption is that tokens that are *less* similar (has larger distance) to others are more important, as their information is *rarer* within the sequence. Let $\mathbf{S} \in \mathbb{R}^{T \times T}$, where $\mathbf{S}_{ij} = \|\mathbf{Z}_{:,i} - \mathbf{Z}_{:,j}\|_2^2$, denote the $l2$ distance between $i$-th and $j$-th token features. Formally, the scores are calculated as $\mathbf{R} = \{\sum_j \mathbf{S}_{ij} : 1 \leq i \leq T\}$.

**Attention Concentration (AttnCon).** Several works have shown that some tokens contribute most of the values in attention maps (Zhang et al., 2023). Based on this insight, we compute attention concentration to determine token importance. Specifically, consider a multi-head ($M$ heads) attention example in a given layer. Let $\mathbf{A} \in \mathbb{R}^{M \times T \times T}$ represent the attention probability map, where $\mathbf{A}_{mij}$ denotes the proportion of attention $j$-th token receives from the $i$-th token in the $m$-th head of the attention. Due to the autoregressive nature of LLMs, $\mathbf{A}_{mij} = 0$ for $j > i$. To calculate the attention concentration of the $j$-th token, we sum over the second dimension of $\mathbf{A}$, and further sum the scores of every head together. Specifically, the importance score is calculated as $\mathbf{R} = \{\sum_{m,i} \mathbf{A}_{mij} : 1 \leq j \leq T\}$.

Note that the computed scores can be seamlessly integrated into the GPTQ framework as described in Sec. 4.2 to preserve the efficiency of the algorithm. We select AttnCon as our final strategy due to its superior performance and present the comparison and analysis of all methods in Sec. 5.3.

### 4.4 Dataset Expansion

In our exploration, we find that important tokens tend to be biased toward specific positions. For example, our heuristic methods inherently select tokens from predefined positions. Moreover, AttnCon consistently assigns higher importance to the initial and final tokens, despite not explicitly enforcing this behavior (as shown in Appendix). This bias may lead to inefficiency, as tokens in other positions are significantly overlooked.

To address this, we propose data expansion, a data augmentation technique designed to "shift" tokens within a sequence, ensuring every token can occupy important positions. Specifically, given a token sequence of length $T$ and an expansion factor of $M$, we generate shifted versions of the sequence by offsetting it by $T/M$, $2T/M$, $3T/M$, ..., $(M-1)T/M$. The excessive tokens are then inserted at the beginning of the sequence. This process effectively distributes token importance more evenly, mitigating positional biases and improving

Table 2: Comparison of recent layer-wise quantization approaches with RSQ+ on multiple downstream tasks. We report perplexity for WikiText and accuracy for all other tasks. The model is quantized to 3-bit. The best-performing method across all quantization approaches is highlighted in bold. We denote the standard deviation across three runs as a subscript.

| Method | Wiki | LAMBADA$_{oai}$ | LAMBADA$_{std}$ | WinoGrande | ArcC | ArcE | HellaSwag | PIQA | MMLU | GSM8k | TruthfulQA | Avg |
|---|---|---|---|---|---|---|---|---|---|---|---|---|
| | | | | LLaMA3-8B-Instruct | | | | | | | | |
| Full Model | 8.311 | 71.9 | 65.0 | 71.7 | 56.7 | 79.6 | 75.8 | 78.5 | 65.6 | 79.9 | 51.7 | 69.7 |
| GPTQ | 10.682$_{.04}$ | 50.7$_{2.0}$ | 44.6$_{1.0}$ | 65.7$_{0.7}$ | 35.8$_{1.4}$ | 56.9$_{2.3}$ | 65.8$_{0.2}$ | 70.4$_{1.7}$ | 50.8$_{0.4}$ | 27.5$_{1.5}$ | 44.8$_{1.0}$ | 51.3$_{0.7}$ |
| QuaRot | 9.517$_{.11}$ | 68.8$_{1.2}$ | 59.7$_{1.0}$ | 70.2$_{1.2}$ | 49.6$_{1.5}$ | 74.9$_{0.9}$ | 71.3$_{0.0}$ | 76.8$_{0.7}$ | 57.6$_{0.5}$ | 61.2$_{1.7}$ | 48.0$_{0.9}$ | 63.8$_{0.5}$ |
| **RSQ+** | **9.046**$_{.01}$ | **70.8**$_{0.3}$ | **62.3**$_{0.4}$ | **70.6**$_{0.7}$ | **50.3**$_{1.1}$ | **76.5**$_{1.3}$ | **72.1**$_{0.2}$ | **77.1**$_{0.5}$ | **60.0**$_{0.3}$ | **63.4**$_{2.4}$ | **50.6**$_{1.7}$ | **65.4**$_{0.1}$ |
| | | | | Mistral-NeMo-12B | | | | | | | | |
| Full Model | 6.095 | 75.8 | 68.3 | 75.1 | 59.1 | 80.1 | 82.2 | 82.1 | 68.2 | 80.8 | 54.8 | 72.7 |
| GPTQ | 9.537$_{.01}$ | 44.6$_{0.6}$ | 29.8$_{4.3}$ | 57.3$_{1.1}$ | 38.8$_{0.2}$ | 58.0$_{1.6}$ | 60.5$_{2.0}$ | 71.2$_{0.6}$ | 48.0$_{1.1}$ | 37.6$_{1.5}$ | 47.0$_{1.5}$ | 49.3$_{0.4}$ |
| QuaRot | 6.782$_{.00}$ | **75.6**$_{0.3}$ | 60.9$_{5.8}$ | 72.8$_{0.7}$ | **55.8**$_{1.5}$ | 76.9$_{1.1}$ | 78.1$_{0.3}$ | 79.7$_{0.4}$ | 63.5$_{0.0}$ | **71.1**$_{1.0}$ | 52.5$_{0.9}$ | 68.7$_{1.1}$ |
| **RSQ+** | **6.673**$_{.01}$ | 75.4$_{0.3}$ | **66.5**$_{0.6}$ | **73.5**$_{0.4}$ | 55.7$_{0.5}$ | **77.2**$_{1.1}$ | **78.5**$_{0.1}$ | **80.7**$_{0.2}$ | **64.3**$_{0.3}$ | **71.1**$_{0.4}$ | **52.9**$_{0.5}$ | **69.6**$_{0.2}$ |
| | | | | Qwen-2.5-7B-Instruct | | | | | | | | |
| Full Model | 5.335 | 75.2 | 68.6 | 72.7 | 59.0 | 76.4 | 85.2 | 81.0 | 83.3 | 84.4 | 65.5 | 75.1 |
| GPTQ | 9.577$_{.03}$ | 53.6$_{1.0}$ | 48.6$_{1.8}$ | 62.1$_{0.9}$ | 47.1$_{1.8}$ | 66.9$_{2.1}$ | 72.8$_{0.0}$ | 73.7$_{0.5}$ | 60.2$_{0.5}$ | 40.9$_{10.5}$ | 53.7$_{1.5}$ | 58.0$_{1.8}$ |
| QuaRot | 8.053$_{.02}$ | 67.9$_{0.3}$ | 62.1$_{0.4}$ | 68.3$_{1.2}$ | **54.0**$_{1.6}$ | **78.7**$_{2.9}$ | 76.7$_{0.4}$ | **78.8**$_{0.8}$ | 68.7$_{0.6}$ | 76.4$_{0.7}$ | 60.0$_{1.4}$ | 69.2$_{0.5}$ |
| **RSQ+** | **8.051**$_{.01}$ | **68.7**$_{0.2}$ | **63.5**$_{0.5}$ | **68.5**$_{0.6}$ | 53.7$_{1.7}$ | 78.2$_{2.0}$ | **77.1**$_{0.3}$ | **78.8**$_{1.0}$ | **69.0**$_{0.4}$ | **77.1**$_{0.8}$ | **61.2**$_{0.9}$ | **69.6**$_{0.3}$ |

overall token utilization. We illustrate this approach in Fig. 1. We denote the variants with and without dataset expansion as RSQ+ and RSQ, respectively.

## 5 Experiments

### 5.1 General Experimental Setups

Our experiments span diverse model families, tasks, and, in some cases, different setups to ensure the generalizability of our approach. However, if not further specified, we use LLaMA3-8B-Instruct as the base model and WikiText-2 as the calibration dataset, consisting of 256 samples with 4096 tokens each. We adopt this model for its long-context capability, which is one focus of our evaluation. We mainly focus on weight-only quantization to highlight the contribution of our method, which modifies the objective in learning quantized weights. More results beyond weight-only quantization (*e.g.* joint quantization of weights, activations, and KV cache) are provided in Sec. 5.5. We mostly quantize models to 3 bits, and evaluate the performance on downstream tasks in zero- or few-shot manners without fine-tuning. **Our experiments are conducted using three different seeds**.

We mainly use GPTQ and QuaRot as the baselines. GPTQ performs layer-wise quantization as described in Sec. 3.1. QuaRot, on the other hand, mitigates outliers by applying rotations (detailed in Sec. 3.2), followed by applying GPTQ to quantize the rotated model. We also demonstrate the compatibility of our approach with state-of-the-art (SOTA) vector quantization approaches in Sec. 5.5. For RSQ, we use AttnCon as the scaling strategy, and set the data expansion factor $M$ to 8 for RSQ+.

### 5.2 Comparison of RSQ Against Baselines

**Setup.** We evaluate RSQ and other baselines across three model families: LLaMA3-8B-Instruct (Dubey et al., 2024), Mistral-NeMo-12B (Jiang et al., 2023), and Qwen-2.5-7B-Instruct (Yang et al., 2024). The quantized models are tested on a diverse set of tasks, including LAMBADA (with two splits: the original paper's version and OpenAI's version) (Paperno et al., 2016), WinoGrande (Sakaguchi et al., 2019) and ARC (Challenge and Easy splits) (Clark et al., 2018), HellaSwag (Zellers et al., 2019), PIQA (Bisk et al., 2020), MMLU (Hendrycks et al., 2021), GSM8k (Cobbe et al., 2021), and TruthfulQA (Lin et al., 2021). Accuracy is used as the evaluation metric.

**Results.** Tab. 2 presents the evaluation results for the 16-bit and 3-bit quantized models using GPTQ, QuaRot, and RSQ+. GPTQ demonstrates a notable performance gap compared to QuaRot and RSQ+, primarily due to the negative impact of outliers in the model, which degrade the quantization quality. When

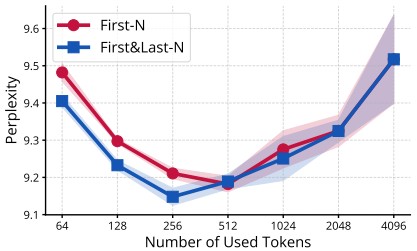
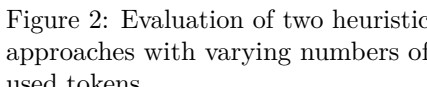

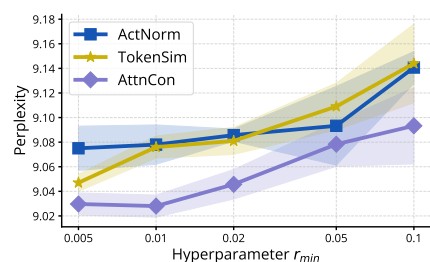

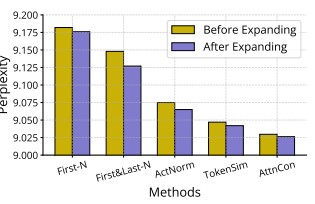

Figure 2: Evaluation of two heuristic approaches with varying numbers of used tokens.

Figure 3: Evaluation of three dynamic approaches with varying $r_{min}$.

Figure 4: The effect of expanding the dataset on different methods.

comparing RSQ+ to QuaRot, our approach achieves superior results in Wiki Perplexity and most evaluation tasks across the three models, where RSQ+ achieves absolute improvements of 1.6%, 0.9%, and 0.4% in average accuracy against QuaRot, respectively. This shows that incorporating token importance improves quantization, even with the same number of calibration tokens. Additional results on other models and quantization cost analysis are in the Appendix.

## 5.3 Evaluating Design Choices in RSQ

**Setup.** We follow the same setup described in Sec. 5.1 for this section. **To avoid overfitting, we only use the perplexity on WikiText-2 as the evaluation metric** in this part, and apply the finalized design across diverse setups to demonstrate its generalizability, as shown in other sections.

**Results.** We first compare the performance of two heuristic approaches, First-N and First&Last-N, across different numbers of activated tokens. As shown in Fig. 2, we observe that perplexity decreases steadily as the number of tokens is reduced from 4096 to around 512 or 256, but increases when using fewer tokens for both approaches. This suggests that **using the fewest tokens does not necessarily yield the best results**. We hypothesize that the model requires a certain number of tokens to effectively capture token interactions that are brought by the attention mechanism. We also observe that First&Last-N often outperforms First-N when using the same number of tokens, achieving their optimal perplexity values of 9.15 and 9.18, respectively. This suggests that **learning is more effective when incorporating the last chunk of tokens into the first chunk, rather than using the first and middle chunks**. In Fig. 3, we compare three dynamic approaches, ActNorm, TokenSim and AttnCon, with varying hyperparameter $r_{min}$ across $\{0.005, 0.01, 0.02, 0.05, 0.1\}$. Among these approaches, we find AttnCon reaches its optimal perplexity (9.028) at $r_{min} = 0.01$ while ActNorm and TokenSim reach their optimal perplexity (9.075, 9.047, respectively) at $r_{min} = 0.005$. These relatively small $r_{min}$ indicate that **placing lower score on less important tokens and focusing more on the important tokens is beneficial**. We adopt AttnCon as our final scaling strategy, as it achieves the best perplexity performance. We also assess the efficacy of data expansion ($M = 8$) by incorporating it into each approach using its optimal hyperparameter. As shown in Fig. 4, most scaling strategies benefit from data expansion in terms of perplexity, but **token scaling still primarily drives the performance gains of RSQ**. More results on data augmentation are in Appendix.

## 5.4 Evaluation on Long-Context Tasks

**Setup.** We use LLaMA3-8B-Instruct as the backbone model for this experiment. Since this model has a context length limit of 8k tokens, the dataset samples used for evaluation do not exceed this length. When evaluating long-context tasks, a natural question arises: *does using longer input sequences in the calibration dataset improve long-context performance?* To explore this, we test three different configurations of the calibration dataset: we set the number of samples to 256, 512, and 1024, with corresponding sequence lengths of 4096, 2048, and 1024 tokens, respectively. Note that we adjust the number of samples to ensure the total number of tokens remained consistent across configurations.

Table 3: Comparison of RSQ+ and QuaRot across multiple long-context benchmarks using three different calibration dataset configurations. The model is quantized to 3-bit. The best-performing method among all quantization approaches is highlighted in bold.

| Method | LITM | | | | L-Eval | | | | | | | | LongICLBench | | |
|---|---|---|---|---|---|---|---|---|---|---|---|---|---|---|---|
| | $P$=1 | $P$=15 | $P$=30 | Avg | TOEFL | QuALITY | Coursera | SFiction | GSM | CodeU | TopicRet | Avg | Banking77 | TecRED | Avg |
| Full Model | 53.67 | 45.61 | 46.33 | 48.54 | 81.04 | 60.40 | 52.62 | 71.88 | 81.00 | 4.44 | 64.67 | 59.43 | 59.40 | 41.65 | 50.52 |
| | | | | | | number of samples = 256, sequence length = 4096 | | | | | | | | | |
| QuaRot | $50.80_{1.6}$ | $43.47_{2.4}$ | $43.29_{2.5}$ | $45.85_{2.0}$ | $72.24_{0.9}$ | $52.15_{0.8}$ | $46.90_{0.7}$ | $\mathbf{65.62_{3.1}}$ | $63.33_{6.6}$ | $2.22_{0.9}$ | $45.11_{3.5}$ | $49.65_{1.4}$ | $\mathbf{62.40_{4.3}}$ | $44.76_{5.2}$ | $53.58_{3.2}$ |
| **RSQ+** | $\mathbf{52.23_{0.5}}$ | $\mathbf{46.03_{2.7}}$ | $\mathbf{45.59_{3.5}}$ | $\mathbf{47.95_{2.0}}$ | $\mathbf{76.21_{2.1}}$ | $\mathbf{54.46_{0.7}}$ | $\mathbf{49.27_{1.3}}$ | $63.80_{4.0}$ | $\mathbf{66.33_{3.8}}$ | $\mathbf{4.07_{0.5}}$ | $\mathbf{50.89_{2.4}}$ | $\mathbf{52.14_{0.3}}$ | $58.60_{7.7}$ | $\mathbf{49.83_{0.7}}$ | $\mathbf{54.21_{3.5}}$ |
| | | | | | | number of samples = 512, sequence length = 2048 | | | | | | | | | |
| QuaRot | $49.17_{1.2}$ | $44.32_{1.0}$ | $43.51_{0.5}$ | $45.67_{0.2}$ | $71.13_{2.3}$ | $51.16_{1.4}$ | $46.95_{0.7}$ | $58.33_{0.9}$ | $65.67_{4.9}$ | $\mathbf{4.81_{2.1}}$ | $48.00_{7.0}$ | $49.43_{2.0}$ | $\mathbf{65.40_{0.6}}$ | $45.48_{3.1}$ | $\mathbf{55.44_{1.8}}$ |
| **RSQ+** | $\mathbf{51.39_{1.6}}$ | $\mathbf{45.89_{0.8}}$ | $\mathbf{45.87_{1.2}}$ | $\mathbf{47.72_{0.8}}$ | $\mathbf{73.23_{2.2}}$ | $\mathbf{53.96_{1.7}}$ | $\mathbf{48.79_{1.0}}$ | $\mathbf{61.98_{2.6}}$ | $\mathbf{70.00_{2.1}}$ | $4.44_{0.0}$ | $\mathbf{51.78_{7.0}}$ | $\mathbf{52.02_{1.9}}$ | $62.33_{2.2}$ | $\mathbf{46.17_{5.5}}$ | $54.25_{3.8}$ |
| | | | | | | number of samples = 1024, sequence length = 1024 | | | | | | | | | |
| QuaRot | $48.66_{2.6}$ | $47.20_{0.9}$ | $\mathbf{48.80_{1.7}}$ | $48.22_{1.4}$ | $73.11_{2.1}$ | $51.82_{1.4}$ | $48.64_{1.8}$ | $62.24_{1.3}$ | $65.00_{0.8}$ | $2.59_{1.3}$ | $40.45_{3.0}$ | $49.12_{0.9}$ | $59.73_{8.0}$ | $46.02_{7.0}$ | $52.87_{6.9}$ |
| **RSQ+** | $\mathbf{51.10_{0.2}}$ | $\mathbf{48.01_{0.7}}$ | $48.62_{1.3}$ | $\mathbf{49.24_{0.2}}$ | $\mathbf{74.72_{2.8}}$ | $\mathbf{55.78_{2.0}}$ | $\mathbf{50.58_{0.8}}$ | $\mathbf{64.85_{3.3}}$ | $\mathbf{69.33_{1.2}}$ | $\mathbf{3.70_{0.5}}$ | $\mathbf{47.55_{7.4}}$ | $\mathbf{52.35_{2.0}}$ | $\mathbf{61.20_{5.2}}$ | $\mathbf{46.30_{1.9}}$ | $\mathbf{53.75_{3.5}}$ |

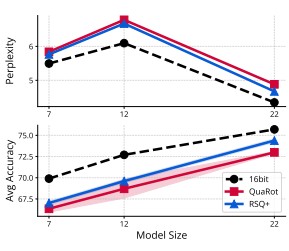

Figure 5: Ablation on model sizes.

Table 4: The ablation study on the calibration datasets.

| Metric | Method | Calibration Dataset | | | |
|---|---|---|---|---|---|
| | | Wiki | RedPajama | C4 | PTB |
| Wiki PPL ↓ | QuaRot | $9.34_{.03}$ | $9.77_{.09}$ | $9.90_{.02}$ | $9.98_{.06}$ |
| | **RSQ+** | $\mathbf{9.00_{.01}}$ | $\mathbf{9.31_{.03}}$ | $\mathbf{9.41_{.03}}$ | $\mathbf{9.61_{.08}}$ |
| Avg Acc (%) ↑ | QuaRot | $64.1_{0.2}$ | $64.2_{0.3}$ | $64.0_{0.2}$ | $63.7_{0.2}$ |
| | **RSQ+** | $\mathbf{65.1_{0.2}}$ | $\mathbf{65.5_{0.3}}$ | $\mathbf{65.6_{0.3}}$ | $\mathbf{64.5_{0.1}}$ |

Table 5: The Ablation study on the number of bits.

| Metric | Method | Number of Bits | | |
|---|---|---|---|---|
| | | 4 | 3 | 2 |
| Wiki PPL ↓ | QuaRot | $8.57_{.06}$ | $9.34_{.03}$ | $22.71_{0.5}$ |
| | **RSQ+** | $\mathbf{8.47_{.01}}$ | $\mathbf{9.00_{.01}}$ | $\mathbf{16.26_{.12}}$ |
| Avg Acc (%) ↑ | QuaRot | $68.1_{0.0}$ | $64.1_{0.2}$ | $35.3_{0.8}$ |
| | **RSQ+** | $\mathbf{68.3_{0.1}}$ | $\mathbf{65.1_{0.2}}$ | $\mathbf{40.6_{0.3}}$ |

Lost in the Middle (LITM) (Liu et al., 2024a) is a retrieval task designed to assess positional biases of the answer placed in the input documents. We set $P = 1, 15, 30$ to indicate that the answer appears in the $P$-th document out of a total of 30 documents. Next, we adopt some closed-ended tasks from L-Eval (An et al., 2024) for evaluating the long-context understanding, such as TOEFL (Tseng et al., 2016; Chung et al., 2018), QuALITY (Pang et al., 2022), Coursera, SFiction, GSM, CodeU, and TopicRet (Li et al., 2023). LongICLBench (Li et al., 2024b) focuses on evaluating the long-context in-context learning capabilities of LLMs. From this benchmark, we sample two datasets: Banking77 (Casanueva et al., 2020) and TecRED (Zhang et al., 2017). Most of the datasets employ accuracy as evaluation metrics, except we use the F1 score for TecRED. More datasets and results are presented in Appendix.

**Results.** Tab. 3 displays our evaluation of long-context tasks using three different calibration dataset configurations. RSQ+ consistently outperforms QuaRot across nearly all benchmarks in all three configurations. This demonstrates that the strategy of prioritizing important tokens produces quantized models that perform effectively on both short- and long-context tasks. Furthermore, the results indicate that **focusing on a subset of tokens is sufficient to capture the long-term dependencies across tokens**. We do not observe a clear trend indicating that using a calibration dataset with longer sequences results in better or worse performance on long-context tasks. **This suggests that simply matching the length distribution between the calibration dataset and downstream tasks is insufficient**. A more advanced strategy, whether through improving the data or the method sides, is needed to further enhance performance in long-context scenarios.

## 5.5 Generalizability of RSQ

**Different Model Sizes.** We choose three models from the mistral family: Mistral-7B-Instruct-v0.3, Mistral-NeMo-12B and Mistral-Small-Instruct-2409, whose sizes are 7B, 12B, and 22B, respectively. We quantize each model to 3-bit. The results (Fig. 5) show that RSQ+ consistently outperforms QuaRot across all three models, with the performance gap being slightly larger in the 22B model. We also apply RSQ to a larger model, LLaMA3-70B-Instruct, to evaluate its quantization results at 2, 3, and 4 bits. As shown in Tab. 6,

Table 6: Comparison RSQ against QuaRot on multiple downstream tasks using LLaMA3-70B-Instruct. Results are run with seed 0.

| Method | Wiki | LAMBADA$_{oai}$ | LAMBADA$_{std}$ | WinoGrande | ArcC | ArcE | HellaSwag | PIQA | MMLU | GSM8k | TruthfulQA | Avg |
|---|---|---|---|---|---|---|---|---|---|---|---|---|
| Full Model | 5.399 | 76.1 | 72.4 | 76.2 | 64.5 | 84.6 | 82.5 | 81.9 | 79.8 | 92.1 | 61.6 | 77.17 |
| | | | | | 4-bit | | | | | | | |
| QuaRot | 5.882 | **76.0** | 71.9 | 75.4 | 64.1 | **84.5** | 81.7 | **82.0** | 77.8 | 91.1 | **61.8** | 76.63 |
| **RSQ** | **5.730** | **76.0** | **72.4** | **75.8** | **64.2** | 84.3 | **81.9** | 81.7 | **79.0** | **91.5** | 60.5 | **76.73** |
| | | | | | 3-bit | | | | | | | |
| QuaRot | 7.511 | 74.2 | 69.0 | 72.3 | 59.2 | 80.9 | 78.4 | 79.2 | 70.4 | 85.1 | 57.2 | 72.59 |
| **RSQ** | **6.503** | **76.8** | **71.7** | **75.5** | **62.1** | **82.9** | **80.9** | **81.0** | **76.0** | **89.8** | **60.8** | **75.75** |
| | | | | | 2-bit | | | | | | | |
| QuaRot | 26.17 | 34.7 | 24.7 | 52.3 | 27.7 | 49.2 | 41.6 | 62.5 | 26.5 | 2.1 | 40.4 | 36.17 |
| **RSQ** | **10.623** | **70.2** | **60.2** | **63.5** | **51.0** | **76.5** | **69.8** | **75.7** | **56.7** | **48.2** | **46.6** | **61.84 ↑** |

RSQ (without data expansion) consistently outperforms QuaRot, particularly at lower bitwidths (2 and 3 bits). This demonstrates that token importance is especially beneficial for extreme model compression.

**Different Calibration Datasets.** In addition to using WikiText-2 as the calibration dataset (Bandari et al., 2024; Ji et al., 2024), we also evaluate RSQ's performance with using RedPajama (Weber et al., 2024), C4 (Raffel et al., 2020), and PTB (Marcus et al., 1993). This and next experiments are done in a legacy setup, where we use 512 data samples, each with 2048 tokens for calibration. The results, presented in Tab. 4, demonstrate that our approach consistently outperforms QuaRot across varying calibration datasets, demonstrating the robustness of the approach.

**Different Bit Precisions.** In previous experiments, we consistently quantized the model to 3-bit. Here, we extend the analysis by exploring the effects of RSQ when quantizing to 2-bit and 4-bit. When quantizing LLaMA3-8B-Instruct with WikiText-2 (512 data samples, each containing 2048 tokens), our results in Tab. 5 demonstrate that RSQ+ outperforms QuaRot across different bit precisions. Notably, the performance gap is larger at lower bit precisions, suggesting that the idea of "learning from important tokens" can be a crucial factor for achieving effective extreme compression. Furthermore, we conduct a 1.58-bit (ternary) (Ma et al., 2024) experiment on LLaMA3-70B-Instruct, and we show that the RSQ also attains a lower wiki perplexity compared to QuaRot (15.25 vs 99.02). We also evaluated a 1-bit post-training setting; however, models quantized by either method failed to achieve meaningful perplexity.

**RSQ in Joint Quantization.** While most of our experiments focus on weight-only quantization, we also apply RSQ to joint quantization schemes: W4A4KV16 and W4A4KV4, which simultaneously quantize weights and activations, and weights, activations, and KV cache, respectively. We follow the implementation used in QuaRot, where activations and KV caches are rotated with the weights to reduce outliers. Next, these components are then quantized using a round-to-nearest approach. The results are presented in Tab. 7, demonstrating that RSQ outperforms QuaRot not only in weight-only quantization but also in the joint quantization setting.

Table 7: Joint quantization.

| Method | Tasks | |
|---|---|---|
| | Wiki PPL ↓ | Avg Acc (%) ↑ |
| | W4A4KV16 | |
| QuaRot | $10.26_{.05}$ | $63.7_{0.1}$ |
| **RSQ** | $\mathbf{10.03}_{.04}$ | $\mathbf{64.3}_{0.1}$ |
| | W4A4KV4 | |
| QuaRot | $12.72_{.16}$ | $57.6_{0.7}$ |
| **RSQ** | $\mathbf{12.32}_{.20}$ | $\mathbf{58.3}_{0.6}$ |

**RSQ with SOTA vector quantization.** In previous experiments, we quantize model weights individually using scalar quantization. In this experiment, we extend RSQ to vector quantization (VQ), which better approximate the high-dimensional weight distribution. Specifically, we incorporate token importance to the SOTA VQ methods, QuIP# and QTIP, and compare their performance against the original versions. Since both already use rotation to address outliers, our modification focuses solely on token importance scaling. We follow their default setup:

Table 8: RSQ + SOTA 2-bit VQ.

| Method | Tasks | | |
|---|---|---|---|
| | C4 PPL ↓ | Wiki PPL ↓ | Avg Acc (%) ↑ |
| QuIP# | 11.62 | 9.029 | 54.92 |
| **RSQ (w/ QuIP#)** | **8.622** | **6.589** | **60.69** |
| QTIP | 9.037 | 6.864 | 62.38 |
| **RSQ (w/ QTIP)** | **8.167** | **6.244** | **65.70** |

2-bit weight quantization on the LLaMA-2-7B (Touvron et al., 2023) model using 256 samples (each with

Table 9: The data augmentation ablation study.

| Method | Wiki PPL | Avg Acc (%) |
|---|---|---|
| QuaRot | $9.517_{.11}$ | $63.8_{0.5}$ |
| QuaRot+ $(M = 8)$ | $9.435_{.04}$ | $64.2_{0.3}$ |
| RSQ | $9.037_{.02}$ | $65.1_{0.1}$ |
| RSQ+ $(M = 8)$ | $9.020_{.01}$ | $65.4_{0.2}$ |
| RSQ (8x data) | $9.012_{.02}$ | $65.5_{0.4}$ |

Table 10: Memory requirements and runtime of RSQ variants and QuaRot.

| Method | Runtime Per Layer (s) | GPU Memory (GB) | RAM (GB) |
|---|---|---|---|
| QuaRot | 63.89 | 24.58 | 17.66 |
| RSQ (w/ ActNorm) | 82.91 | 24.58 | 17.67 |
| RSQ (w/ AttnCon) | 114.41 | 24.58 | 17.67 |
| RSQ+ (w/ CPU offloading) | 847.06 | 8.15 | 214.15 |

4096 tokens) from RedPajama, and evaluation on C4, WikiText-2, and five downstream tasks (ARC Easy, ARC Challenge, BoolQ (Clark et al., 2019), PIQA, and WinoGrande). We disable their model and layer fine-tuning to isolate quantization effects. Notably, we reuse scaling strategies and hyperparameters searched in Sec. 5.3 without retuning, and no data expansion is applied. We present the results in Tab. 8, where it shows that RSQ greatly boosts the accuracy and can be seamlessly applied to SOTA VQ methods.

## 5.6 Data Augmentation vs More Data

We develop the data augmentation based on the observation that LLMs treat some positions more important than others, so we shift the tokens to enable better utilization of tokens and we demonstrate this strategy can improve the performance. Data augmentation trades compute for performance, offering an alternative to collecting more data, which often requires additional effort. While collecting more data can be useful to directly increase the number of tokens, but it doesn't share the same insight as our data augmentation for position bias. Moreover, collecting text data can be non-trivial and also involve many filtering and cleaning steps (Dubey et al., 2024; DeepSeek-AI, 2024).

To further evaluate the effectiveness of data augmentation, we compare it against directly using $8\times$ the calibration data, as shown in Tab. 9. The results indicate that both strategies improve performance, but data augmentation achieves comparable gains with significantly less real data.

We also apply data augmentation to QuaRot, and refer to this variant as QuaRot+. The results in Tab. 9 show that data augmentation benefits both methods, but RSQ still outperforms QuaRot+ by a clear margin even without augmentation, highlighting that RSQ's improvements stem primarily from its core idea on token importance, not the dataset size.

## 5.7 Inference Efficiency and Quantization Efficiency

Our key contribution lies in modifying the GPTQ objective to incorporate token importance, and this affects only the quantization process without altering the quantized weight format or inference kernels. This design makes our approach a plug-and-play module that seamlessly leverages existing kernels for fast inference and memory efficiency, while enhancing the quantitative performance. As a result, we achieve the "exact same" speedup and memory savings as the underlying frameworks we build upon (e.g., GPTQ, QuIP#, QTIP, as referenced in this paper).

The following is the detail for quantization cost analysis for the AttnCon approach. First, for each layer, we forward the cached inputs (hidden states for the data from calibration dataset) through this layer to obtain the layer's outputs and the Hessian for the weight matrices within this layer. For AttnCon, we compute $(QK^\top)V$ (ignoring softmax/normalization as they do not affect complexity), then sum the attention to get token importance. With $Q, K, V \in \mathbb{R}^{L \times D}$, the cost for $QK^\top$ is $\mathcal{O}(L^2D)$, for $(QK^\top)V$ is $\mathcal{O}(L^2D)$, and for summation along the token dimension is $\mathcal{O}(L^2)$.

Hence, token importance only adds small computation cost, $\mathcal{O}(L^2)$, on top of the attention cost $\mathcal{O}(L^2D)$, which is already computed to obtain the layer outputs and Hessian. However, we need to switch from SDPA attention to standard attention to obtain attention maps, which incurs some additional computation.

As shown in Tab. 10, the GPU memory and RAM usage for different approaches show negligible differences when data augmentation is not utilized, and this setup already achieves strong performance as shown in Tab. 9

(QuaRot vs RSQ). Additionally, quantization of each layer with RSQ can be completed in approximately 1 - 2 minutes, where the runtime overhead comes from the above analysis. Lastly, as aforementioned, our data augmentation serves as an advanced technique that trades efficiency (memory and runtime) for accuracy.

# 6 Conclusion

This paper begins with the observation that focusing on important tokens can lead to better quantization performance than using all tokens uniformly. This insight motivates the development of RSQ (Rotate, Scale, then Quantize) and the exploration of various scaling strategies. The effectiveness and generalizability of RSQ are validated across a wide range of benchmarks and configurations.

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

## A   Transformer Architecture

The core components of a transformer are the attention layers and feed-forward network (FFN). An attention layer consists of four linear modules: $\mathbf{W}_q$, $\mathbf{W}_k$, $\mathbf{W}_v$ and $\mathbf{W}_o$. Similarly, an FFN contains three linear modules: $\mathbf{W}_{up}$, $\mathbf{W}_{gate}$, $\mathbf{W}_{down}$. Attention layers are responsible for capturing token dependency and global information while FFNs, positioned after each attention block, perform token-wise feature transformations.

## B   A Global Loss Perspective on Quantization with Token Importance

One perspective to understand tokens are not equally important is through the lens of connecting local reconstruction loss to the model's global (final) loss. Ideally, we could optimize quantization by directly minimizing the global loss $\mathcal{L}_g$ with respect to the quantized weights in a given layer $l$:

$$\mathcal{L}_g(\hat{\mathbf{W}}^{(l)}) - \mathcal{L}_g(\mathbf{W}^{(l)}) \tag{5}$$

This difference can be approximated by the product of the activation change and the corresponding gradient, assuming the weight perturbation from quantization is small:

$$\mathcal{L}_g(\hat{\mathbf{W}}^{(l)}) - \mathcal{L}_g(\mathbf{W}^{(l)}) \approx \frac{\partial \mathcal{L}}{\partial \mathbf{X}^{(l+1)}} \cdot \delta \mathbf{X}^{(l+1)} = \frac{\partial \mathcal{L}}{\partial \mathbf{X}^{(l+1)}} \cdot (\hat{\mathbf{W}}^{(l)}\mathbf{X}^{(l)} - \mathbf{W}^{(l)}\mathbf{X}^{(l)}) \tag{6}$$

Taking the squared norm on both sides gives us a closer link between global loss change and the layer reconstruction loss used in GPTQ:

$$||\mathcal{L}_g(\hat{\mathbf{W}}^{(l)}) - \mathcal{L}_g(\mathbf{W}^{(l)})||_2^2 \approx ||\frac{\partial \mathcal{L}}{\partial \mathbf{X}^{(l+1)}} \cdot (\hat{\mathbf{W}}^{(l)}\mathbf{X}^{(l)} - \mathbf{W}^{(l)}\mathbf{X}^{(l)})||_2^2 \tag{7}$$

$$= \sum_{i=0}^{T} \sum_{j=0}^{dout} ||\frac{\partial \mathcal{L}}{\partial \mathbf{X}^{(l+1)}}|_{j,i} \times (\hat{\mathbf{W}}^{(l)}\mathbf{X}^{(l)} - \mathbf{W}^{(l)}\mathbf{X}^{(l)})_{j,i}||_2^2 \tag{8}$$

Here, the activation gradient $\frac{\partial \mathcal{L}}{\partial \mathbf{X}^{(l+1)}}$ can be interpreted as reweighting the layer reconstruction loss. This gradient has the same shape as the activation $\mathbf{X}^{(l+1)}$, i.e., $\mathbb{R}^{dout \times T}$ (output dimension $\times$ sequence length) for a single sample. Notably, computing and storing these activation gradients is significantly more expensive than standard layer-wise quantization. For instance, consider a 32-layer model with 7 linear weights per layer, each of size $4096 \times 4096$. To store the gradients for one sample, we require: $32(\text{layers}) \times 7(\text{weights}) \times 4096(\text{output dimension}) \times 4096(\text{sequence length}) \times 2(\text{2 bytes if we store in 16 bit})/10^9 \approx 7.5(\text{GB})$. Storing this information for a 256-sample calibration set would be prohibitively expensive, especially for larger models. Even if offloaded to disk, the storage remain substantial.

Thus, we further assume that the importance can be decomposed along the channel and token dimensions:

$$\sum_{i=0}^{T} \sum_{j=0}^{dout} ||\frac{\partial \mathcal{L}}{\partial \mathbf{X}^{(l+1)}}|_{j,i} \times (\hat{\mathbf{W}}^{(l)}\mathbf{X}^{(l)} - \mathbf{W}^{(l)}\mathbf{X}^{(l)})_{j,i}||_2^2 \tag{9}$$

$$\approx \sum_{i=0}^{T} \sum_{j=0}^{dout} || \mathbf{r}_{\mathbf{token}_i} \times \mathbf{r}_{\mathbf{channel}_j} \times (\hat{\mathbf{W}}^{(l)}\mathbf{X}^{(l)} - \mathbf{W}^{(l)}\mathbf{X}^{(l)})_{j,i}||_2^2 \tag{10}$$

$$\overset{\text{if GPTQ}}{=} \sum_{i=0}^{T} \sum_{j=0}^{dout} || \mathbf{r}_{\mathbf{token}_i} \times (\hat{\mathbf{W}}^{(l)}\mathbf{X}^{(l)} - \mathbf{W}^{(l)}\mathbf{X}^{(l)})_{j,i}||_2^2 \tag{11}$$

The final equality holds due to a limitation in GPTQ, which quantizes each row (i.e., output dimension) independently, meaning reweighting across output dimensions does not affect the quantization result. However, applying reweighting across tokens can affect the outcome and thus offers a viable mechanism to reduce the global loss via token-level importance. Therefore, we justify our layer-wise objective and the intuition that tokens are not equally important through the lens of the global loss. One way to represent token importance is through the norm of the activation gradient over the output dimension, expressed as $\mathbf{r}_{\mathbf{token}_i} = \sqrt{\sum_{j=0}^{dout}(\frac{\partial \mathcal{L}}{\partial \mathbf{X}^{(l+1)}})_{j,i}^2}$. This formulation is significantly more storage-efficient, as the score becomes 1-dimensional after aggregating over the channel dimension.

However, computing token importance via activation gradients still requires backpropagation through the entire model, making it inefficient. To avoid this overhead, we instead use lightweight proxies for token importance that are likely indicative of their overall contribution to the final loss. For instance, we prioritize tokens with high attention weights (AttnCon) or large activation norms (ActNorm), as such tokens have greater influence on other tokens, have a larger impact on subsequent layer outputs, and may ultimately contribute more to the final loss. These strategies maintain the fully layer-wise nature of our method while still aligning with the objective of minimizing global loss.

We compare our primary scaling strategy, AttnCon, with the activation gradient-based method, termed ActGrad. In ActGrad, we precompute per-sample activation gradients on the original model (LLaMA3-8B-Instruct) using two A6000 GPUs, convert them into token scores as described above, and then use these scores to reweight tokens during quantization. We use the cross-entropy loss on the next tokens as our global loss. We report WikiText-2 perplexity results using both the raw gradient norms and their mapped versions using our mapping function in Fig. 7. The results show that AttnCon performs competitively with ActGrad while being significantly more efficient. We hypothesize that ActGrad's performance may be limited because the activation gradients are computed on the original model and do not reflect changes introduced during layer-wise quantization.

## C  Extented Task Details

### C.1  Downstream Task Details

The quantized models are tested on a diverse set of tasks, including LAMBADA (with two splits: the original paper's version and OpenAI's version) (Paperno et al., 2016) for word prediction, WinoGrande (Sakaguchi et al., 2019) and ARC (Challenge and Easy splits) (Clark et al., 2018) for commonsense reasoning, HellaSwag (Zellers et al., 2019) for commonsense natural language inference, PIQA (Bisk et al., 2020) for physical commonsense reasoning, MMLU (Hendrycks et al., 2021) as a comprehensive knowledge benchmark, GSM8k (Cobbe et al., 2021) for grade school math, and TruthfulQA (Lin et al., 2021) to assess the model's truthfulness in generation. All datasets use accuracy as the evaluation metric.

We perform 5-shot and 8-shot evaluation for MMLU and for GSM8k, respectively. All other tasks are evaluated in a zero-shot setting, following the defaults specified in: `https://github.com/EleutherAI/lm-evaluation-harness/tree/main/lm_eval/tasks`.

### C.2  Long-context Benchmark Details

Lost in the Middle (LITM) (Liu et al., 2024a) is a retrieval task designed to assess positional biases of the answer placed in the input documents. We set $P = 1, 15, 30$ to indicate that the answer appears in the $P$-th document out of a total of 30 documents (avg length=4.5k). LongEval dataset (Li et al., 2023) includes a synthetic retrieval task, where each line consists of a key-value pair. Given an input sample of $L$ lines, the model is asked to extract the value corresponding to a specified key in the query. The hyperparameter $L$ controls both the input length and the task complexity. We set $L = 300, 460$, and $620$, where the corresponding input lengths are around 4k, 6k, and 8k, respectively. Next, we adopt a couple of closed-ended tasks from L-Eval (An et al., 2024) for evaluating the long-context understanding. TOEFL (avg length=3.6k) (Tseng et al., 2016; Chung et al., 2018), QuALITY (6.2k) (Pang et al., 2022), and Coursera (6.8k) are multiple-choice QA tasks, while SFiction (7.2k) is a True/False QA task. GSM (4.8k) evaluates the model's in-context learning ability, CodeU (7.4k) assesses its capability to deduce program outputs, and TopicRet (7.6k) (Li et al., 2023) is designed as a retrieval task. LongICLBench (Li et al., 2024b) focuses on evaluating the long-context in-context learning capabilities of LLMs. From this benchmark, we sample two datasets: Banking77 (avg length=7.7k) (Casanueva et al., 2020) and TecRED (6.6k) (Zhang et al., 2017). These datasets primarily test the model's ability to learn and generalize a large number of concepts using only a few-shot examples. LongCodeArena (Bogomolov et al., 2024) is a benchmark for code processing tasks that go beyond a single file and require project-wide context. We sample the library-based code generation task to evaluate the ability of the model to solve tasks by using a given library.

Most of the datasets employ accuracy as evaluation metrics, except we use the F1 score for TecRED and ChrF (Popović, 2015) for library code generation in LongCodeArena.

## D  Extented Related Work

Recent state-of-the-art open-source LLMs (Yang et al., 2024; Dubey et al., 2024; Jiang et al., 2024) typically exceed 50 billion parameters. While their released weights make them accessible, their massive weight size poses significant challenges for practitioners and limits the feasibility of fine-tuning or even inference with

these models. Pruning and (Frantar & Alistarh, 2023; Sun et al., 2024b; Sung et al., 2024) and quantization are common strategies to compress these models' weight, and this paper focuses on quantization as it is often more effective (Kuzmin et al., 2023).

Post-training quantization (PTQ) has gained significant attention for its ability to quantize pre-trained models without requiring expensive retraining. Methods such as ZeroQuant (Yao et al., 2022) and QLoRA (Dettmers et al., 2023) apply round-to-nearest techniques to the weights, even without utilizing calibration datasets. While these approaches are extremely efficient, they often yield suboptimal performance due to their lack of information about the data distribution. To address this limitation, several PTQ methods leverage calibration datasets for improved quantization. These data-dependent PTQ methods are also often referred to as layer-wise quantization methods, as they typically quantize models one layer at a time for efficiency. Specifically, GPTQ (Frantar et al., 2023) and OBC (Frantar & Alistarh, 2022) quantize weights and adjust the remaining weights with data-derived Hessian matrices (second-order statistics) accordingly. AWQ (Lin et al., 2024b) minimizes quantization error by rescaling weights based on the activation distribution to protect important "channels", where RSQ modifies the layer reconstruction objective to prioritize important "tokens". The two methods are conceptually complementary (channel importance vs token importance), and may be composable. SqueezeLLM (Kim et al., 2024) utilizes sensitivity-based k-means clustering to learn a quantization grid. QuIP# (Tseng et al., 2024), AQLM (Egiazarian et al., 2024) and QTIP (Tseng et al., 2025) represent groups of quantized weights with vectors instead of scalar values. Additionally, several approaches focus on fine-tuning parameters introduced during quantization (*e.g.*, quantization indices, group scales, and group zeros), such as OmniQuant (Shao et al., 2024), HQQ (Badri & Shaji, 2023), and PV-Tuning (Malinovskii et al., 2024).

One challenge in weight quantization is the presence of outliers in the weights, as LLMs often contain a subset of weights with exceptionally large magnitudes (Dettmers et al., 2024; Kim et al., 2024; Yu et al., 2024). During quantization, these outliers increase the range of the quantized weights. As a result, most weights are forced into a narrow range to fit the outliers within the quantized representation, which ultimately leads to suboptimal performance. One way to address this challenge is through mixed-precision quantization (Dettmers et al., 2024; 2022; Kim et al., 2024); however, current hardware lacks efficient support for inference using this technique. Recent studies, such as QuIP (Chee et al., 2023) and QuaRot (Ashkboos et al., 2024b), have empirically shown that outliers in weights can be effectively mitigated by applying hadamard orthogonal transformations (hadamard rotations) (Ashkboos et al., 2024a), which reduce their impact and minimize the need for mixed-precision quantization. Furthermore, DuQuant (Lin et al., 2024a) and SpinQuant (Liu et al., 2024b) optimize the rotation matrix through data instead of relying on pre-defined rotations.

Different from previous advancements in layer-wise quantization, RSQ focuses on improving the learning objective of layer-wise quantization methods by prioritizing important tokens. Our approach also builds on QuaRot by applying rotations to mitigate weight outliers and leveraging GPTQ for quantization. This integration makes RSQ a comprehensive and holistic quantization strategy.

# E  Evaluation Details

The section includes the detail of our experiments, and our code also has been released to ensure reproducibilty. All baselines are rerun using their official hyperparameters, on our machine, with the same calibration dataset and three random seeds as our approach, ensuring a fair and consistent comparison across all methods.

**Model.**  As noted in the main paper, we use the instruction-tuned LLaMA3 model for most experiments to enable a unified evaluation of our quantization method on both short- and long-context tasks. Many long-context tasks require strong instruction-following ability, making LLaMA3-Instruct more suitable than the base models used in prior work. Nonetheless, we also present quantization results for LLaMA3-8B-Base and LLaMA2-7B-Base in Tab. 15, showing that RSQ outperforms QuaRot on these models as well. For experiments on QuIP#, QTIP and SpinQuant, we use their default model: LLaMA2-7B-base.

**Calibration Dataset.**  Following prior works (Ashkboos et al., 2024b; Liu et al., 2024b; Shao et al., 2024), we calibrate our model on the training set of WikiText-2 and evaluate perplexity on the validation set. Importantly, we use WikiText-2 results solely to select hyperparameters for the scaling strategy, and report

additional results on downstream tasks and across diverse configurations (model families, calibration datasets, tasks, and quantization methods) using the same hyperparameters to demonstrate the generalizability of our approach. We use 256 examples, each with 4096 tokens, for calibration–longer than the default QuaRot setup (128×2048)–to match the extended 8K context length of our chosen model (LLaMA3) compared to LLaMA2's 4K. Nevertheless, we also show in Tab. 14 that RSQ consistently outperforms QuaRot under the 128×2048 setup. For LLaMA3-70B-Instruct experiments, we reduce the calibration dataset size to 64 samples to fit the quantization process within a single A6000 GPU.

For experiments relying on external repositories such as QuIP#, QTIP, and SpinQuant, we follow their default calibration setup: using 256 samples (each with 4096 tokens) from RedPajama for QuIP# and QTIP, and 128 samples (each with 2048 tokens) from WikiText-2 for SpinQuant. In SpinQuant, we use 800 samples for rotation optimization.

We use longer context length because we adopted LLaMA3 instead of LLaMA2. As you mentioned, QuaRot quantizes the model on the samples with context length of 2048, which is the half of the the context length of LLaMA-2 (=4k). LLaMA-3 has a context length of 8k so we set the context length for the calibration dataset to be 4096 (half of 8k). Note that in Table 3, we also demonstrated that RSQ outperforms QuaRot in different configurations (256x4096, 512x2048, 1024x1024).

**Seeds.** Most of our experiments are run with three seeds: 0, 1, and 2. For experiments on the LLaMA3-70B-Instruct model, we use a single seed (seed = 0). For experiments that rely on external repositories, such as QuIP#, QTIP, and SpinQuant, we follow their default setup and report results from a single run using seed 0.

**Experiment Compute Resources.** Our approach is built on a layer-wise framework, making it memory-efficient. RSQ can quantize most models using a single A6000 GPU (48GB). For 70B models, we reduce the calibration size to 64 samples as noted above to fit within memory constraints. Due to the expanded calibration data size, we offload the model's hidden states to CPU for RSQ+ reduce GPU memory usage. The concrete memory usage for quantizing LLaMA3-8B-Instruct with RSQ variants and QuaRot is shown in Tab. 10, which is measured on a single A6000 GPU and an Intel(R) Xeon(R) Silver 4416+ CPU. The table shows that incorporating scaling does not increase the memory requirements (with some runtime overhead), while data augmentation enables a trade-off between computation and performance.

In this paper, we follow QuaRot and use "fake quantization," where weights are quantized to the target bitwidth and then cast to 16-bit, resulting in no inference speed-up during experiments. However, as shown in the main paper, RSQ is compatible with various quantization methods and can leverage their optimized kernels for efficient inference. We adopt fake quantization to isolate and highlight our core contribution: token importance for quantization. In our experiments, we use a single A6000 GPU for all short-context tasks when the model size is ≤22B. For long-context tasks, LLaMA3-8B-Instruct is also evaluated on a single A6000. For Qwen2.5-32B, we use one A100 (80GB) and reduce the number of shots to 4 for GSM8k. For LLaMA3-70B-Instruct, we use 3×A100 GPUs for most tasks and 8×A100 for GSM8k.

## F   Additional Evaluation Results and Analysis

### F.1   Additional Results on Long-Context Evaluation

We evaluate the quantized LLaMA3 on the LongEval and LongCodeArena benchmark (details in Sec. C.2) and present the results in Tab. 11. Our findings indicate a clear performance improvement with RSQ compared to QuaRot. Moreover, in the LongEval evaluation, we observe a larger performance drop as input lengths increase, which aligns with the findings reported by Li et al. (2024a).

In the LITM evaluation, we observe that LLaMA3-8B-Instruct generally performs better when the answer appears in the early documents ($P = 1$). However, its performance on later documents does not consistently surpass that on middle documents, deviating from the findings reported by Liu et al. (2024a) for other models. Interestingly, we also find that the quantized models outperform the 16-bit model by 2–4% in average accuracy on LongICLBench, similar to the findings from Hong et al. (2024). This shows that quantization does not

Table 11: Comparison of RSQ+ and QuaRot on LongEval and LongCodeArena tasks. The model is quantized to 3-bit. The best-performing method among all quantization approaches is highlighted in bold.

| Method | LongEval | | | | LongCodeArena |
|---|---|---|---|---|---|
| | $L$=300 | $L$=460 | $L$=620 | Avg | CodeGen |
| Full Model | 100 | 98.80 | 82.00 | 93.60 | 0.298 |
| number of samples = 256, sequence length = 4096 | | | | | |
| QuaRot | $99.47_{0.3}$ | $90.87_{2.6}$ | $52.80_{6.0}$ | $81.04_{2.9}$ | $0.206_{.013}$ |
| RSQ+ | $\mathbf{99.60}_{0.2}$ | $\mathbf{94.67}_{3.9}$ | $\mathbf{58.00}_{10.3}$ | $\mathbf{84.09}_{4.8}$ | $\mathbf{0.231}_{.008}$ |
| number of samples = 512, sequence length = 2048 | | | | | |
| QuaRot | $99.67_{0.1}$ | $90.87_{2.7}$ | $54.47_{3.5}$ | $81.67_{0.2}$ | $0.205_{.007}$ |
| RSQ+ | $\mathbf{99.80}_{0.1}$ | $\mathbf{93.73}_{1.7}$ | $\mathbf{55.33}_{7.1}$ | $\mathbf{82.95}_{2.7}$ | $\mathbf{0.227}_{.002}$ |
| number of samples = 1024, sequence length = 1024 | | | | | |
| QuaRot | $\mathbf{99.80}_{0.2}$ | $85.53_{1.8}$ | $49.60_{2.2}$ | $78.31_{0.9}$ | $0.220_{.001}$ |
| RSQ+ | $99.67_{0.3}$ | $\mathbf{88.33}_{6.3}$ | $\mathbf{52.00}_{12.9}$ | $\mathbf{80.00}_{6.5}$ | $\mathbf{0.225}_{.002}$ |

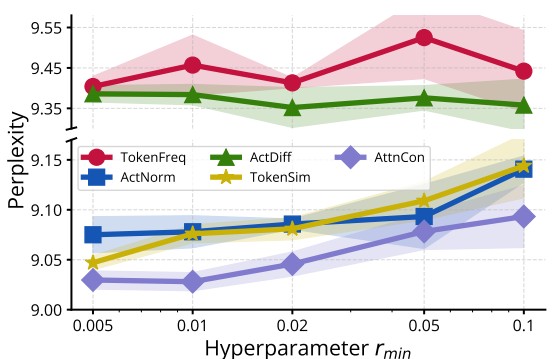

Figure 6: Evaluation of five dynamic approaches with varying $r_{min}$.

Figure 7: Comparison between AttnCon and ActGrad (reach 9.036 using original values).

necessarily hurt performance in every aspect and can yield improvements in certain scenarios. One possible explanation is that quantization might reduce weight noise, potentially enhancing the model's robustness for some tasks.

### F.2 Additional Dynamic Scaling Strategies

We present the two additional dynamic strategies in this section.

**Token Frequency (TokenFreq).** This approach assumes that a token's importance is related to its frequency, and we observe that assigning greater weight to *less* frequent tokens yields better results. We compute token frequency based on the calibration dataset used for quantization, denoting the occurrence count of token t as $C(\mathrm{t}) \in \mathbb{R}$. Given the input token sequence $\{\mathrm{t}_1, \mathrm{t}_2, ..., \mathrm{t}_T\}$, we define token importance $\mathbf{R} \in \mathbb{R}^T$ as $\{-C(\mathrm{t}_i) : 1 \leq i \leq T\}$.

**Activation Difference (ActDiff).** Our next approach defines token importance based on the feature changes between inputs and outputs (Sajjad et al., 2023). We observe that assigning greater weight to tokens with *smaller* changes yields better results compared to assigning greater weight to those with larger changes. This suggests that these "steady tokens" play a more crucial role in the model. Specifically, the scores are calculated as $\mathbf{R} = \{-\|\mathtt{Layer}(\mathbf{Z}_{:,i}) - \mathbf{Z}_{:,i}\| : 1 \leq i \leq T\}$.

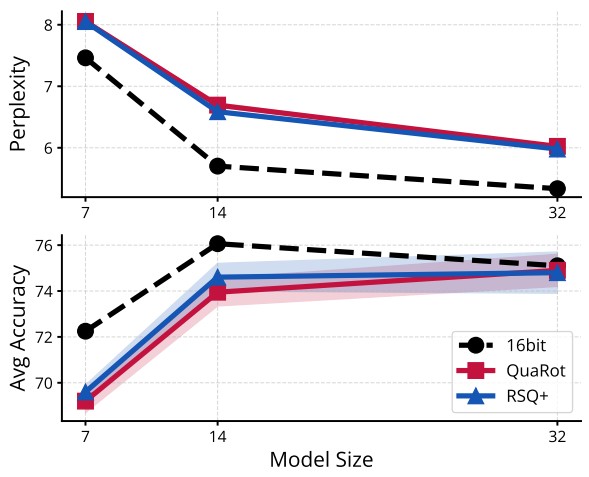

Figure 8: Ablation on model sizes using Qwen2.5.

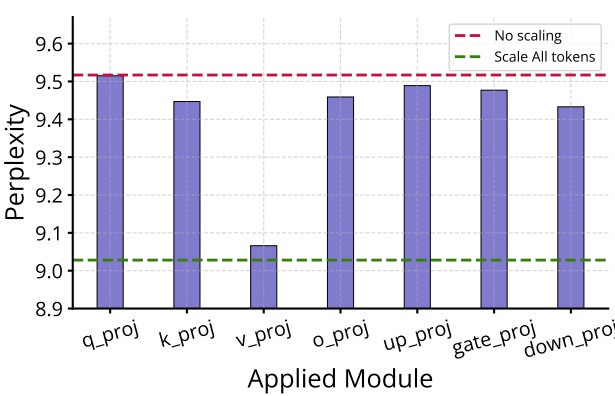

Figure 9: Ablation results on WikiText-2 about quantizing different modules with RSQ.

Their evaluation results on WikiText-2 perplexity using LLaMA3-8B-Instruct are shown in Fig. 6, where we observe that TokenFreq and ActDiff perform less competitively than the other approaches, suggesting that token frequency and feature changes after a layer are not strong indicators of token importance for quantizing LLMs.

### F.3 Scaling Model Sizes for Qwen2.5

We also demonstrate RSQ's and QuaRot performance on different sizes of Qwen2.5 modes (7B, 14B, and 32B) in Fig. 8, showing that RSQ still outperform the baseline for the three models.

### F.4 Applying RSQ on each module independently

Generally, we apply RSQ to all transformer modules simultaneously, including query, key, and value projection layers in attention layers and up, gate, and down projection layers in feed-forward networks. In this section, we conduct an ablation study where RSQ is applied independently to each module, while the remaining modules use "uniform" token scaling (i.e., no scaling). The results, presented in Fig. 9, indicate that while most modules benefit from RSQ, the most significant improvement is observed in `v_proj`. We hypothesize that this is because the values (outputs of `v_proj`) have the most direct influence on all other tokens compared to other modules. However, we leave a deeper exploration of this phenomenon for future work.

### F.5 Wiki Eval with Different Context Lengths

Perplexity on WikiText is a widely used metric for evaluating quantization approaches. However, the context length of the evaluation set can significantly impact perplexity values, and previous studies have employed different setups. For instance, for LLaMA models, AQLM (Egiazarian et al., 2024) reports results with a context length of 4096, while QuaRot (Ashkboos et al., 2024b) uses 2048.

While we use a context length of 2048 for most of the experiments in this paper, we also report WikiText perplexity at additional context lengths (512 and 8192) in Fig. 10 for three models. We observe that the performance gap between different approaches remains relatively consistent across context lengths, suggesting that either of the lengths can serve as a reliable metric for comparing methods. However, as expected, longer contexts generally lead to lower perplexity, likely because having more preceding tokens for LLMs to attend to improves prediction accuracy. Therefore, we emphasize that when using perplexity as a comparison metric, it is crucial to maintain consistent settings to ensure fair and meaningful comparisons.

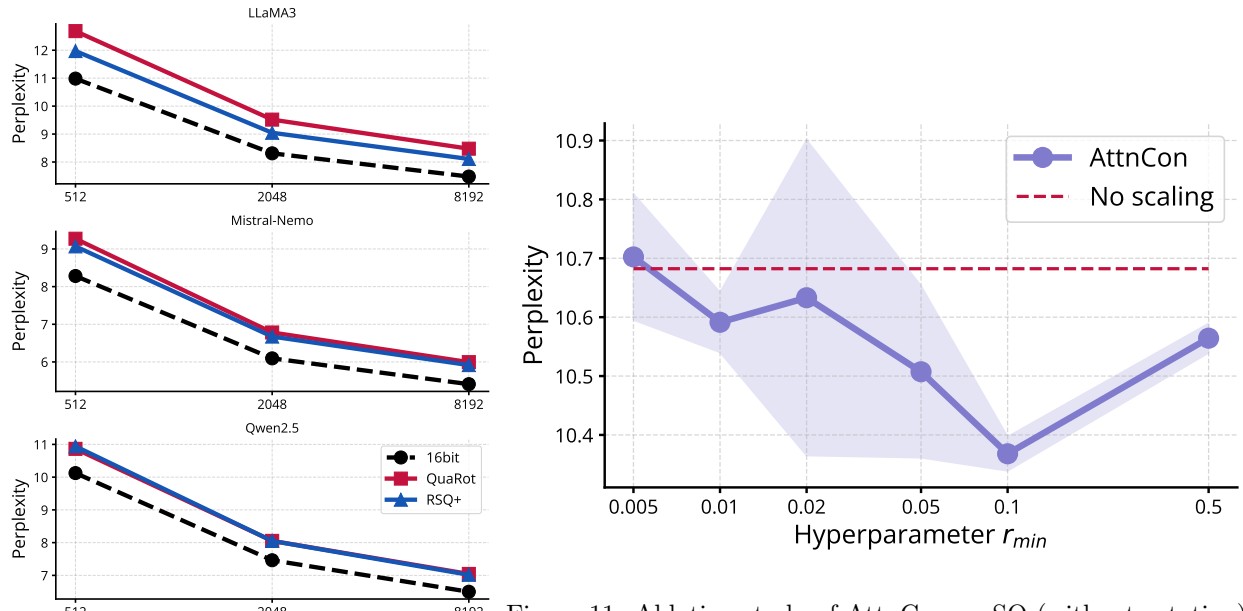

Figure 11: Ablation study of AttnCon on SQ (without rotation).

Figure 10: WikiText evaluation with three different context lengths for three models.

Table 12: RSQ + NF.

| Method | Wiki PPL | Avg Acc (%) |
|--------|----------|-------------|
| QuaRot | $9.417_{.06}$ | $63.8_{0.7}$ |
| **RSQ** | $\mathbf{9.047}_{.01}$ | $\mathbf{65.3}_{0.5}$ |

Table 13: RSQ + SpinQuant.

| Method | Wiki PPL | |
|--------|----------|----------|
| | Step=20 | Step=100 |
| QuaRot | 6.222 | 6.852 |
| **RSQ** | **6.160** | **6.818** |

Table 14: Quanitzation using 128 samples with 2048 sequence length.

| Method | Wiki PPL | Avg Acc (%) |
|--------|----------|-------------|
| QuaRot | $9.382_{.02}$ | $64.0_{0.1}$ |
| **RSQ** | $\mathbf{9.054}_{.01}$ | $\mathbf{65.3}_{0.2}$ |

## F.6 The Effect of Scaling without Rotation

In previous experiments, we applied the scaling strategy to weights after rotation. In this section, we investigate its effect when the weights remain unrotated. We refer to this approach as SQ (Scale, then Quantize) and present the results of applying AttnCon on SQ in Fig. 11. Our findings indicate that the best perplexity is achieved at $r_{min} = 0.1$, which is significantly larger than the optimal value observed when weights are rotated ($r_{min} = 0.005$). This suggests that scaling is far more effective when applied to rotated weights. We leave further investigation of this phenomenon for future work.

## F.7 RSQ with Non-uniform Quantization

In the main paper, we have shown that RSQ works compatible with uniform weight quantization and vector quantization. In this subsection, we show that RSQ can also improves the results while using non-uniform quantization grid, such as Normal Float (3-bit) (Dettmers et al., 2023), where we demonstrate the results in Tab. 12.

## F.8 RSQ with SpinQuant

In this subsection, we apply RSQ to the SOTA rotation-based method SpinQuant (Liu et al., 2024b), which optimizes rotation matrices rather than using random hadamard matrices. We follow their original setup with 4-bit quantization for weights, activations, and KV cache on LLaMA2-7B-Base, and evaluate performance on WikiText-2. We test two configurations with rotation optimization steps set to 20 and 100, respectively. As

Table 15: RSQ + LLaMA base models.

| Method | Wiki PPL | Avg Acc (%) |
|--------|----------|-------------|
| LLaMA3-8B-Base | | |
| QuaRot | $7.826_{.04}$ | $59.1_{0.4}$ |
| **RSQ** | $\mathbf{7.118}_{.00}$ | $\mathbf{60.7}_{0.4}$ |
| LLaMA2-7B-Base | | |
| QuaRot | $6.337_{.02}$ | $53.9_{0.5}$ |
| **RSQ** | $\mathbf{5.828}_{.01}$ | $\mathbf{56.2}_{0.4}$ |

shown in Tab. 13, RSQ further improves results when combined with SpinQuant. Interestingly, we observe that fewer optimization steps yield better performance.

### F.9 Experiments on More Models

In this subsection, we show that RSQ also improves performance on base models, such as LLaMA2-Base and LLaMA3-Base, not just instruction-tuned models, as presented in Tab. 15. We also observe a significant performance drop on GSM8k for these base models, suggesting the post-training in the instruction-tuned models is crucial for this math dataset.

## G  Limitation

This paper proposes to perform weight quantization while considering token importance. We introduce several token scaling strategies and conduct extensive experiments across various models and setups to demonstrate the generalizability of the approach. This paper does not present theoretical claims or proofs, but empirically validates the effectiveness of the approach instead. This work is limited to the widely adopted autoregressive LLM architecture and does not explore alternatives such as Mamba or Mixture-of-Experts. Additionally, our study is confined to the language domain for models, datasets, and tasks, and leave investigations in other modalities for future work.

## H  Visualization

We visualize the token importance scores assigned by adaptive approaches for three samples and three layers (the 3rd, 11th, and 21st layers), as shown in Figs. 12 to 16. Note that we clamp the values into the range [0.05-th quantile, 99.95-th quantile] for better visualization.

For TokenFreq (Fig. 12), the scores are close to one for most tokens, suggesting that many tokens in a sequence appear very less frequently. ActDiff (Fig. 14) does not exhibit any clear patterns. In contrast, ActNorm (Fig. 13) reveals that the first token tends to have a slightly larger norm, particularly in the 3rd and 11th layers. For TokenSim (Fig. 15), we observe that the first token is significantly less similar to others in earlier layers but becomes more similar in deeper layers. Lastly, AttnCon (Fig. 16) consistently assigns higher scores to the initial and final tokens across all layers.

## I  Licenses

**Models.**

- LLaMA2: `https://ai.meta.com/llama/license/`

- LLaMA3: `https://www.llama.com/llama3/license/`

- Mistral: `https://github.com/openstack/mistral/blob/master/LICENSE`

- Qwen2.5 `https://huggingface.co/Qwen/Qwen2.5-7B/blob/main/LICENSE`

**Datasets.**

- WikiText-2: `https://huggingface.co/datasets/mindchain/wikitext2`

- C4: `https://huggingface.co/datasets/legacy-datasets/c4`

- RedPajama: `https://github.com/togethercomputer/RedPajama-Data/blob/main/LICENSE`

- PTB: `https://github.com/nlp-compromise/penn-treebank/blob/master/LICENSE`

- LM Eval (for short-context tasks): `https://github.com/EleutherAI/lm-evaluation-harness/blob/main/LICENSE.md`

- Lost in the Middle `https://github.com/nelson-liu/lost-in-the-middle/blob/main/LICENSE`

- L-Eval `https://github.com/OpenLMLab/LEval/blob/main/LICENSE`

- LongICLBench `https://github.com/TIGER-AI-Lab/LongICLBench/blob/main/LICENSE`

- LongEval `https://github.com/DachengLi1/LongChat/blob/longeval/LICENSE`

- LongCodeArena `https://github.com/JetBrains-Research/lca-baselines/blob/main/LICENSE`

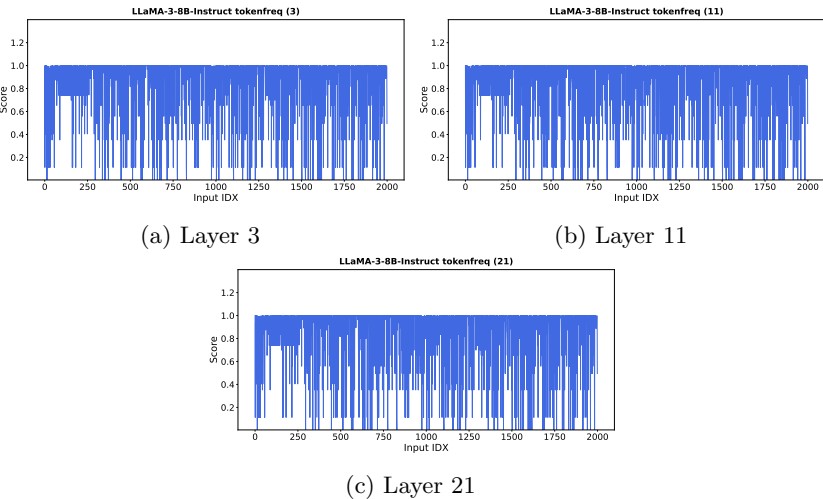

(a) Layer 3      (b) Layer 11

(c) Layer 21

**Figures (a) - (c):** first example.

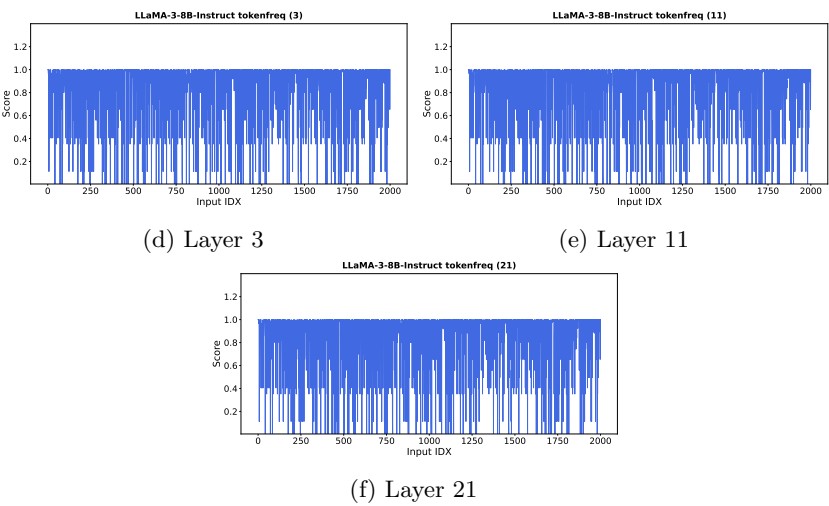

(d) Layer 3      (e) Layer 11

(f) Layer 21

**Figures (d) - (f):** second example.

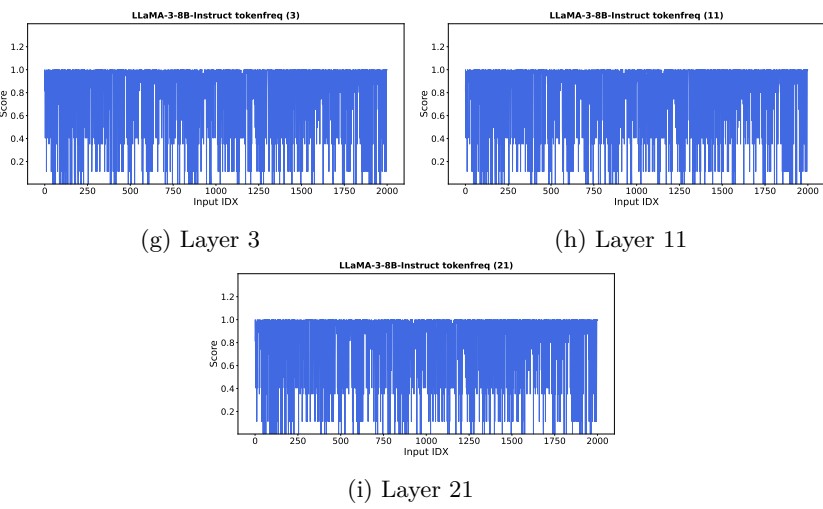

(g) Layer 3      (h) Layer 11

(i) Layer 21

**Figures (g) - (i):** third example.

Figure 12: Visualization of TokenFreq scores across three layers for three different examples.

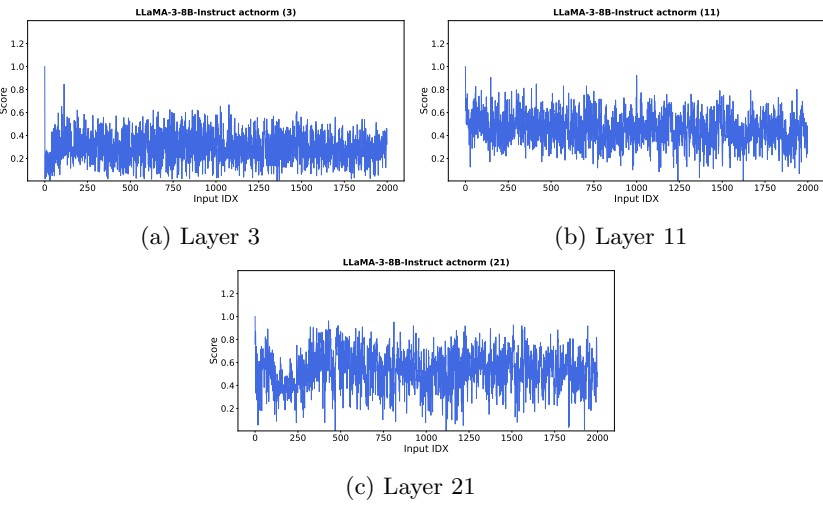

(a) Layer 3        (b) Layer 11

(c) Layer 21

**Figures (a) - (c):** first example.

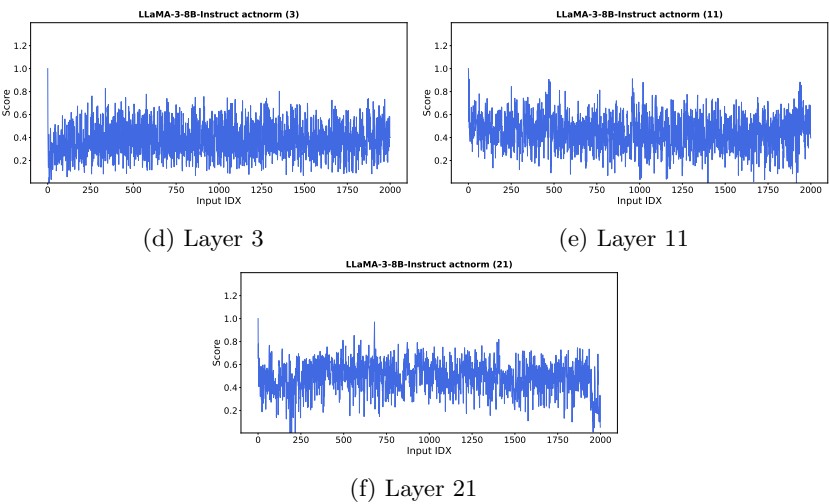

(d) Layer 3        (e) Layer 11

(f) Layer 21

**Figures (d) - (f):** second example.

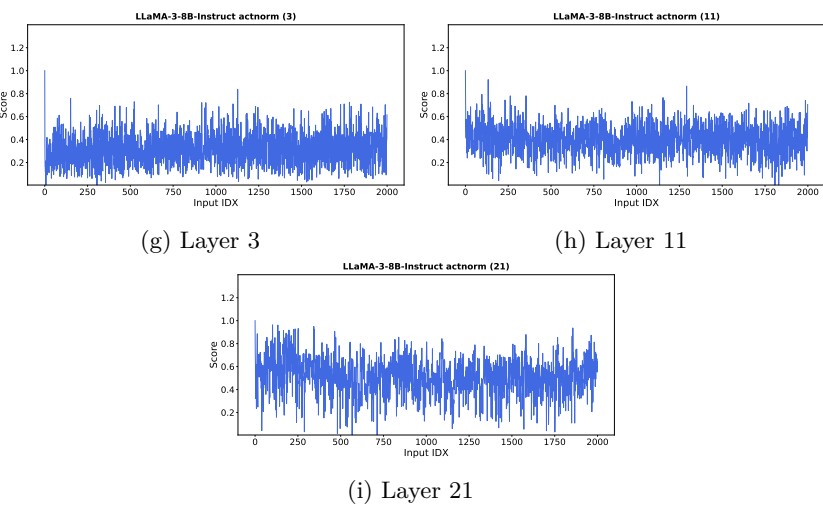

(g) Layer 3        (h) Layer 11

(i) Layer 21

**Figures (g) - (i):** third example.

Figure 13: Visualization of ActNorm scores across three layers for three different examples.

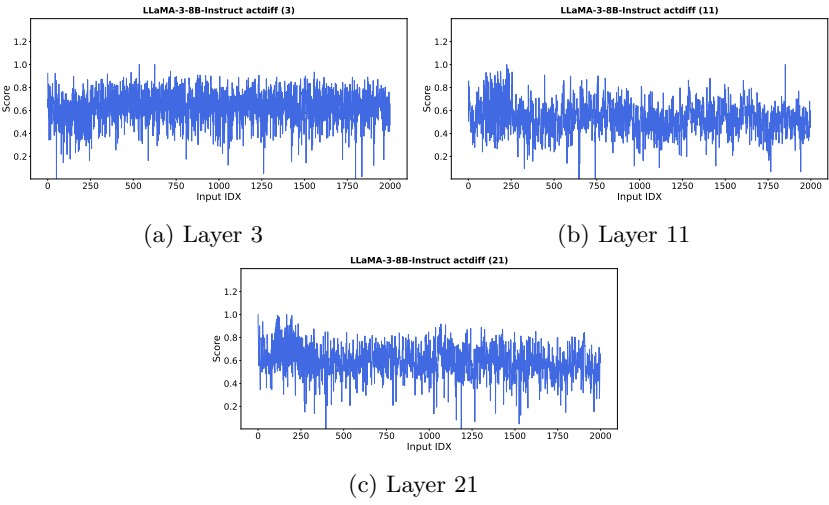

**Figures (a) - (c):** first example.

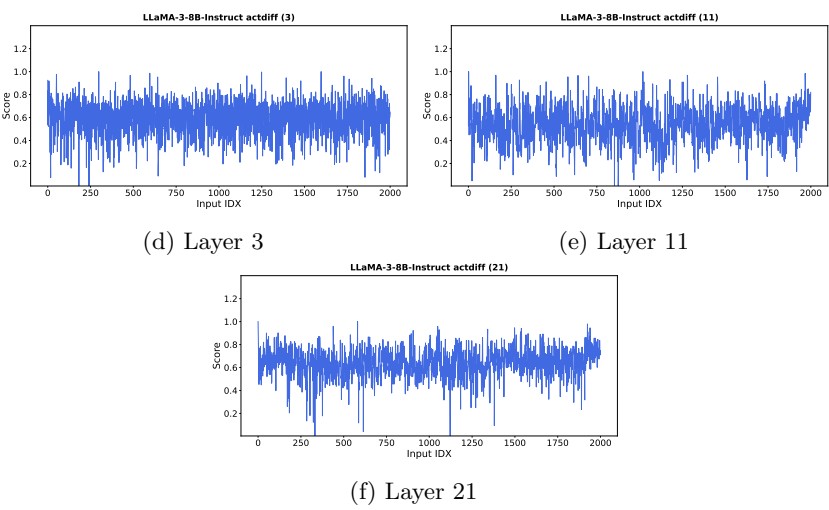

**Figures (d) - (f):** second example.

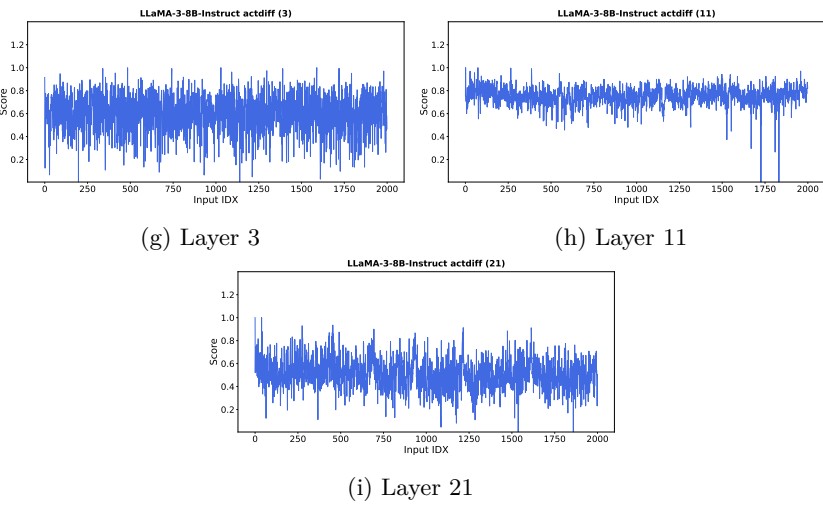

**Figures (g) - (i):** third example.

Figure 14: Visualization of ActDiff scores across three layers for three different examples.

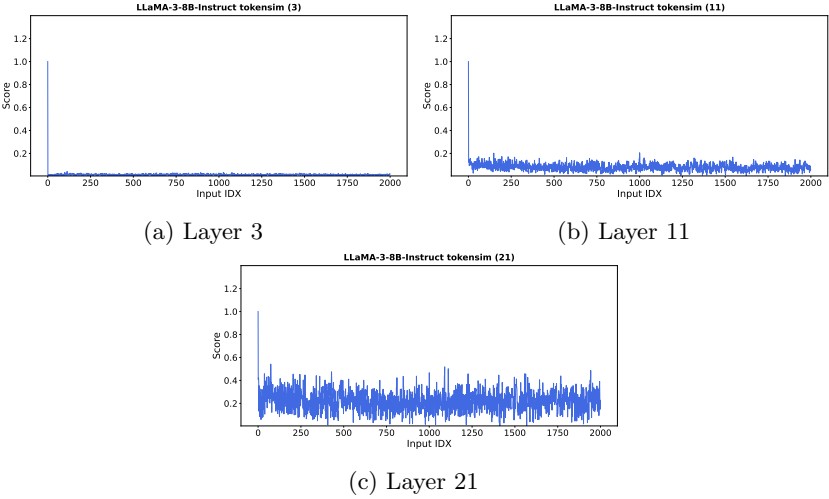

**Figures (a) - (c):** first example.

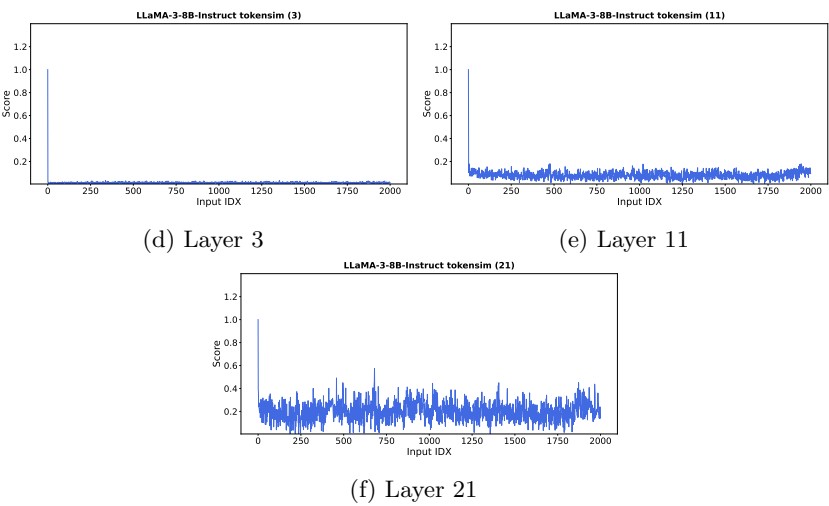

**Figures (d) - (f):** second example.

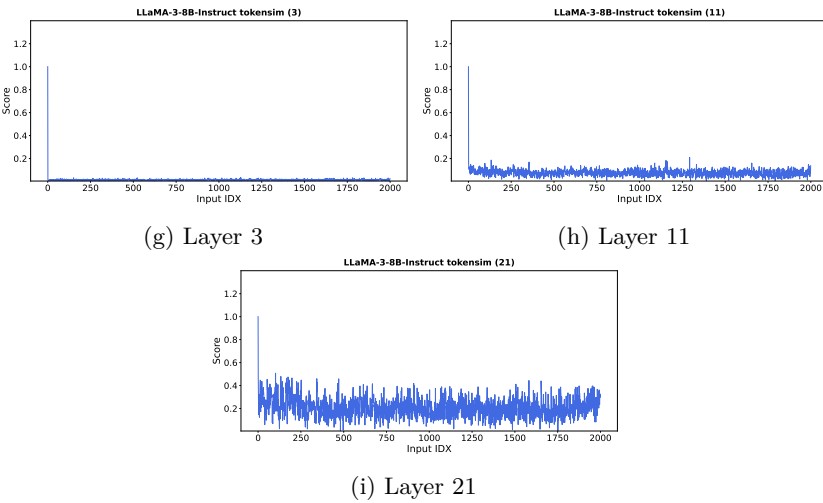

**Figures (g) - (i):** third example.

Figure 15: Visualization of TokenSim scores across three layers for three different examples.

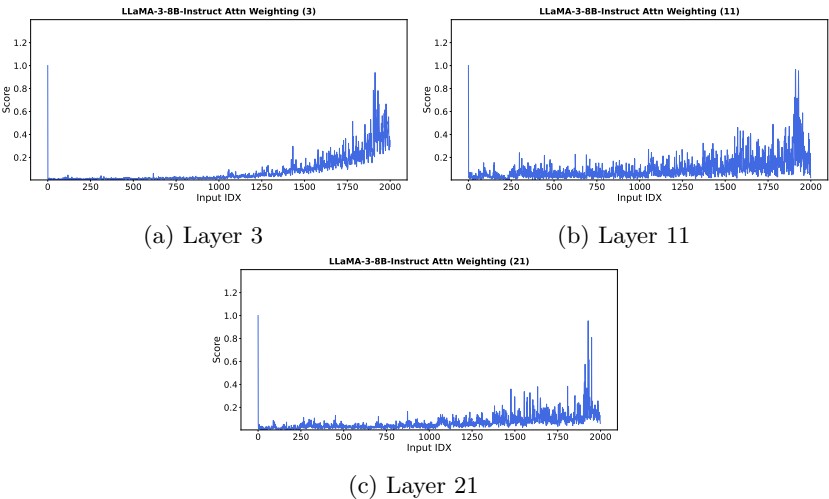

(a) Layer 3          (b) Layer 11

(c) Layer 21

**Figures (a) - (c):** first example.

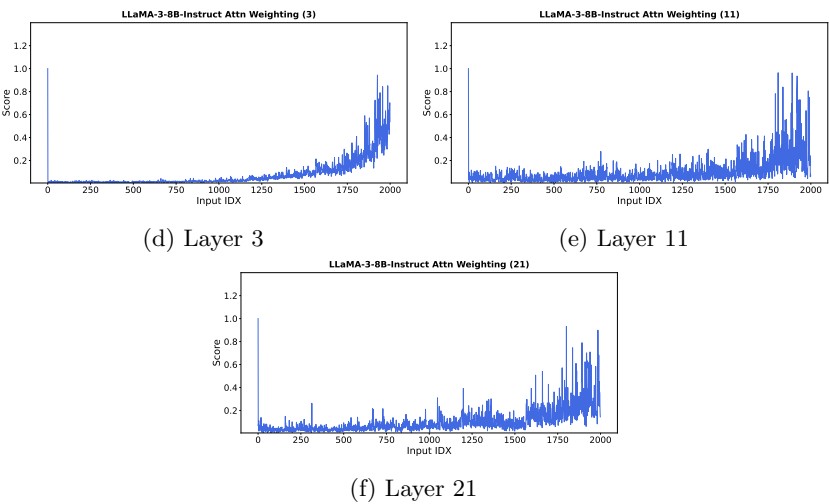

(d) Layer 3          (e) Layer 11

(f) Layer 21

**Figures (d) - (f):** second example.

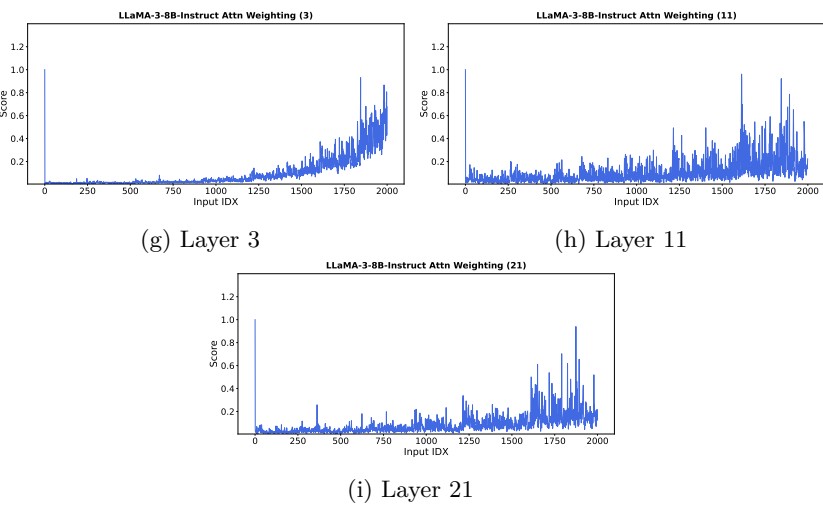

(g) Layer 3          (h) Layer 11

(i) Layer 21

**Figures (g) - (i):** third example.

Figure 16: Visualization of AttnCon scores across three layers for three different examples.