# OpenReview forum: "RSQ: Learning from Important Tokens Leads to Better Quantized LLMs"
_TMLR — Accepted by TMLR_

### Review · Reviewer_9ttG · 2026-02-23

**Summary Of Contributions:**

This paper shows that layer-wise post-training quantization benefits from focusing reconstruction on important tokens rather than treating all tokens equally. It introduces RSQ (Rotate–Scale–Quantize), which weights token activations by importance, estimated from attention concentration, and integrates this weighting into the GPTQ-style second-order objective, while remaining compatible with existing PTQ pipelines. Across multiple LLM families and evaluation suites (including long-context benchmarks), RSQ consistently improves low-bit quantization quality, especially at 2–3 bits.

**Audience:**

Yes

**Audience Explanation:**

Yes. Many in TMLR’s audience working on efficient LLM deployment, model compression, and post-training quantization would find the paper’s findings useful, since it offers a simple, pipeline-compatible way to improve low-bit PTQ by leveraging token importance.

**Broader Impact Concerns:**

Not applicable to this paper.

**Claims And Evidence:**

Yes

**Claims Explanation:**

Mostly yes. This paper provides broad empirical evidence across multiple model families, bit-widths, calibration settings, and both standard and long-context benchmarks, with consistent gains over strong PTQ baselines. The ablations on token-importance strategies and the RSQ+ shift augmentation further strengthen the causal story. That said, the mechanistic explanation for why attention-concentration importance is optimal is still largely empirical, and the added computation cost for AttnCon could be characterized more transparently.

**Requested Changes:**

In my view, this paper presents solid and careful work, and I only have a few minor questions that I hope the authors can clarify. First, in the AttnCon description in Section 4.3, the text provides only a summation formula; I would appreciate more implementation details, such as which layer(s) and head(s) are used in practice. Do you aggregate over all attention heads, and do you apply any additional normalization? Second, while AWQ is mentioned in the Related Work, the paper does not elaborate on how it differs from RSQ or whether the two are composable. If the scope of this work is intended to focus specifically on GPTQ-like methods rather than a general PTQ improvement paradigm, it would be helpful to state this explicitly in the paper.

---

> ### Author Response · Authors · 2026-03-27
>
> We sincerely appreciate the reviewers’ time and thoughtful feedback on our submission.
>
> We are encouraged that the reviewers found the paper **promising and solid** [wW8G, 9ttG]. We also appreciate the recognition that the paper provides **extensive empirical validation** [wW8G, WBfj, 9ttG], and that the proposed framework is **compatible with existing PTQ pipelines** and provides **consistent improvements** [wW8G, 9ttG].
>
> The following is our response to your comments.
>
> ---
> > Please provide more implementation details for AttnCon (layers, heads, normalization).
>
> Thank you for requesting this clarification.
>
> As mentioned in Sec 4.3, in practice, during quantization of layer (l), we compute token importance using the attention maps from the **same** layer. Specifically:
>
> - We collect the attention probability tensor ($A \in \mathbb{R}^{M \times T \times T}$), where ($M$) is the number of heads (**all heads are used**).
>
> - For each token (j), we sum the attention probabilities received across all query tokens and all heads:
> $$
> r\_j = \sum\_{m,i} A\_{mij}
> $$
> - The resulting scores are only normalized to the range ([r_{min}, 1]) using the linear transformation described in Eq. (4), not with additional normalization.
>
>
> > The paper does not elaborate on how it differs from RSQ or whether the two are composable.
>
> Thank you for raising this point.
>
> AWQ and RSQ address different sources of quantization error:
> - AWQ rescales weights based on activation distributions to protect important **channels**.
> - RSQ modifies the layer reconstruction objective to prioritize important **tokens**.
>
> Thus, the two methods operate on different dimensions of the quantization problem (channel importance vs token importance).
>
> Furthermore, RSQ is designed as a general objective-level improvement for layer-wise PTQ frameworks, particularly those based on second-order reconstruction (e.g., GPTQ, QuaRot, QuIP#, QTIP). Because AWQ does not use the GPTQ-style Hessian formulation, directly integrating the RSQ objective into AWQ is less straightforward.
>
> Nevertheless, the two ideas are conceptually complementary: channel-level importance (AWQ) and token-level importance (RSQ) could potentially be combined. We will clarify this distinction and discuss possible composability in the related work section, and also revise the scope to improve GPTQ-like methods.

---

### Review · Reviewer_WBfj · 2026-03-04

**Summary Of Contributions:**

The paper introduces RSQ (Rotate, Scale, then Quantize), a post-training quantization (PTQ) framework that prioritizes "important" tokens during layer-wise reconstruction. It combines orthogonal rotations (to mitigate outliers) with a novel token-scaling mechanism based on attention concentration. Key strengths include significant performance gains in low-bit (3-bit) regimes and long-context tasks across multiple model families (LLaMA3, Mistral, Qwen2.5). A potential weakness is the added complexity of the "dataset expansion" (shifting) strategy and the reliance on heuristic-driven hyperparameter tuning for scaling bounds.

**Audience:**

Yes

**Audience Explanation:**

The TMLR audience, particularly those focused on efficient machine learning and LLM compression, will find the shift from "uniform" to "importance-aware" layer-wise optimization highly relevant. As models grow larger, PTQ becomes essential; the insight that attention scores can guide quantization objectives offers a practical and effective refinement to existing frameworks like GPTQ.

**Broader Impact Concerns:**

The work focuses on technical optimization for model compression. There are no immediate ethical concerns. By making LLMs more efficient, it potentially democratizes access to large models on consumer hardware, which generally has a positive social impact. No additional Broader Impact Statement is required.

**Claims And Evidence:**

Yes

**Claims Explanation:**

The authors provide extensive empirical evidence across three distinct model families and multiple benchmarks (WikiText-2, LAMBADA, MMLU, etc.). The "pilot study" clearly demonstrates the non-uniform importance of tokens. Ablation studies on heuristic vs. dynamic strategies (ActNorm, TokenSim, AttnCon) and the "dataset expansion" technique provide a transparent view of the design choices. The consistent improvement over strong baselines like QuaRot and GPTQ, especially in long-context scenarios, validates the core claims.

**Requested Changes:**

To strengthen the work, I recommend the following adjustments:

1. Provide a more formal justification for why the modified Hessian ($H_{RSQ} = 2XR^2X^\top$) remains the optimal second-order statistic under the importance-weighted loss.

2. Hyperparameter Sensitivity: Include a more detailed sensitivity analysis for the $r_{min}$ parameter across different model sizes; currently, it is mostly tuned on LLaMA3-8B.

3. Computational Overhead: Explicitly quantify the additional time/memory cost of the "dataset expansion" and "AttnCon" computation during the calibration phase compared to standard QuaRot.

4. Comparison with AWQ: While GPTQ and QuaRot are addressed, a brief comparison or discussion regarding AWQ (which also uses activation-based scaling) would clarify RSQ's unique positioning.

---

> ### Author Response · Authors · 2026-03-27
>
> We sincerely appreciate the reviewers’ time and thoughtful feedback on our submission.
>
> We are encouraged that the reviewers found the paper **promising and solid** [wW8G, 9ttG]. We also appreciate the recognition that the paper provides **extensive empirical validation** [wW8G, WBfj, 9ttG], and that the proposed framework is **compatible with existing PTQ pipelines** and provides **consistent improvements** [wW8G, 9ttG].
>
> The following is our response to your comments.
>
> ---
>
> > Why the modified Hessian remains optimal under importance-weighted loss
>
> We follow the OBS [1]/GPTQ formulation and derive the Hessian under the importance-weighted reconstruction loss. The following math is based on a weight vector space since rows are processed independently.
>
> Let $w \in \mathbb{R}^{d_{\text{in}}}$ be one row of $W$, and $X = [x_1,\dots,x_T] \in \mathbb{R}^{d_{\text{in}}\times T}$ the calibration features. Let $R = \mathrm{diag}(r_1,\dots,r_T)$ denote token importance.
>
> The RSQ loss between $w$ and $\tilde w$ is:
> $$
> \mathcal{L}(w)
> = \sum_{i=1}^T r_i^2(w^\top x_i - \tilde w^\top x_i)^2.
> $$
> Define $\delta = \tilde w - w$. Then:
> $$
> \mathcal{L}(\delta)
> = \sum_{i=1}^T r_i^2(\delta^\top x_i)^2
> = \delta^\top (X R^2 X^\top)\delta
> = \delta^\top H_{\text{RSQ}}\delta,
> $$
> where $H_{\text{RSQ}} = 2XR^2X^\top$.
> Optimal solution (OBS-style)
> We minimize the loss change when quantizing one weight $w_q$:
> $$
> \min_\delta \tfrac{1}{2}\delta^\top H_{\text{RSQ}}\delta
> \quad
> \text{s.t.}\quad
> e_q^\top \delta = w_q - \tilde w_q.
> $$
> Using a Lagrange multiplier,
>
> $$
> \mathcal J = \tfrac{1}{2}\delta^\top H_{\text{RSQ}}\delta- \lambda(e_q^\top \delta - (w_q - \tilde w_q)),
> $$
> The optimality condition gives
> $$
> H_{\text{RSQ}}\delta + \lambda e_q = 0
> \Rightarrow
> \delta = -\lambda H_{\text{RSQ}}^{-1} e_q.
> $$
> Substitute $\delta$ the constraint,
> $$
> \lambda = -\frac{w_q - \tilde w_q}{[H_{\text{RSQ}}^{-1}]_{qq}},
> $$
> so
> $$
> \delta
> = -\frac{w\_q - \tilde w\_q}{[H\_{\text{RSQ}}^{-1}]\_{qq}} H\_{\text{RSQ}}^{-1} e\_q
> = -\frac{w\_q - \tilde w\_q}{[H\_{\text{RSQ}}^{-1}]\_{qq}} H\_{\text{RSQ},q:}^{-1}.
> $$
>
> This matches GPTQ with $H$ replaced by $H_{\text{RSQ}}$ (in Equation (1) of the paper), showing the modified Hessian is still the optimal second-order statistic under weighted loss.
>
> > Hyperparameter Sensitivity
>
> We thank the reviewer for this suggestion. We evaluate $r_{\min} \in \{0.005, 0.01, 0.02, 0.05, 0.1\}$ on LLaMA3-8B-Instruct across multiple importance strategies, and apply the selected values to all settings. To reduce cost, we do not tune larger models, which may leave minor gains on the table. However, in early experiments on Qwen2.5-7B, we did some searches on hyperparameters, but didn’t find meaningful improvements, and this also prompted us to use shared hyperparameters instead.
>
> > Computational Overhead
>
> We cache layer inputs from calibration data and forward each sample to obtain outputs for Hessian computation to maximize memory efficiency.
>
> For AttnCon, we compute $(QK^\top)V$ (ignoring softmax/normalization as they do not affect complexity), then sum the attention to get token importance. With $Q,K,V \in \mathbb{R}^{L\times D}$:
> - $QK^\top$: $\mathcal{O}(L^2D)$
> - $(QK^\top)V$: $\mathcal{O}(L^2D)$
> - summation along the token dimension: $\mathcal{O}(L^2)$
>
> Token importance adds only $\mathcal{O}(L^2)$ on top of the attention cost $\mathcal{O}(L^2D)$, which is already computed when we compute the Hessian matrix. However, we need to switch from SDPA attention to standard attention to obtain attention maps, which incurs some additional computation.
>
> Dataset expansion increases feature storage and Hessian computation linearly, while GPTQ quantization cost is unchanged.
>
> Empirically (Table 10), RSQ’s memory usage remains similar to QuaRot (~24.6GB). Runtime overhead comes from the above analysis:
> - QuaRot: ~63.9s/layer
> - RSQ (ActNorm): ~82.9s/layer
> - RSQ (AttnCon): ~114.4s/layer
>
> Thus, RSQ adds moderate calibration cost while preserving memory and inference efficiency.
>
> > Comparison with AWQ
>
> AWQ and RSQ address different sources of quantization error:
> - AWQ rescales weights based on activation distributions to protect important **channels**.
> - RSQ modifies the layer reconstruction objective to prioritize important **tokens**.
>
> Thus, the two methods operate on different dimensions of the quantization problem (channel importance vs token importance).
>
> Furthermore, RSQ is designed as a general objective-level improvement for layer-wise PTQ frameworks, particularly those based on second-order reconstruction (e.g., GPTQ, QuaRot, QuIP#, QTIP). Because AWQ does not use the GPTQ-style Hessian formulation, directly integrating the RSQ objective into AWQ is less straightforward.
>
> Nevertheless, the two ideas are conceptually complementary: channel-level importance (AWQ) and token-level importance (RSQ) could potentially be combined. We will clarify this distinction and discuss possible composability in the related work section.

---

### Review · Reviewer_wW8G · 2026-03-13

**Summary Of Contributions:**

The paper proposes a new framework for quantizing LLMs. In particular, it has been empirically observed that some tokens contribute a large fraction of the attention values so existing approaches that consider reconstruction uniformly over all input tokens do not take advantage of this phenomenon. The authors propose a new framework for quantization called RSQ that mitigates the affects of outlier weights via rotation matrices. These rotations causes weights to be more well-conditioned, resulting in better quantization. Experiments provided show the improvements over existing approaches. Crucially, the framework does not require novel quantization algorithms: existing quantization algorithms can be used as a black-box.

**Additional Comments:**

Can we even further take the bit-quantization to the limit for 1.58 or 1 bit models?

**Audience:**

Yes

**Audience Explanation:**

The approach seems promising as (1) it leads to improvements on a variety of LLM architectures and datasets and (2) the method seems generally simple as we can apply existing quantization algorithms as a black-box.

**Claims And Evidence:**

Yes

**Claims Explanation:**

Yes, the methodology seems to be reasonable, and some of the strategies are cited with relevant related works. Additionally, all experiments seem to be reasonable, with setup extensively documented.

**Requested Changes:**

I think the paper is quite strong already. The authors test their framework on a variety of datasets, quantization schemes, and quantization bits. It would be stronger if the authors could test their method on larger language models (in particular more experiments around 70B size). However, this is not critical for my acceptance.

---

> ### Author Response · Authors · 2026-03-27
>
> We sincerely appreciate the reviewers’ time and thoughtful feedback on our submission.
>
> We are encouraged that the reviewers found the paper **promising and solid** [wW8G, 9ttG]. We also appreciate the recognition that the paper provides **extensive empirical validation** [wW8G, WBfj, 9ttG], and that the proposed framework is **compatible with existing PTQ pipelines** and provides **consistent improvements** [wW8G, 9ttG].
>
> The following is our response to your comments.
>
> ---
> > It would be stronger if the authors could test their method on larger language models (in particular more experiments around 70B size).
>
> > Can we even further take the bit-quantization to the limit for 1.58 or 1 bit models?
>
> Thank you for this helpful suggestion. We agree that validating RSQ more on larger models or lower bitness is important.
> Our paper has presented 2-bit results on LLaMA3-70B-Instruct in Table 6, showing a clear improvement from RSQ to QuaRot in terms of wiki perplexity (26.17 -> 10.623) and downstream accuracy (36.17 -> 61.84).
> During the rebuttal, we further conducted a 1.58-bit (ternary) experiment on LLaMA3-70B-Instruct, and we show that the RSQ attains much lower wiki perplexity compared to QuaRot (15.25 vs 99.02). We also tried the 1-bit setting, but unfortunately, the models quantized with both methods do not present a meaningful perplexity.

---

### Decision · Action_Editor_DGEE · 2026-04-19

**Recommendation:** Accept as is

**Additional Comments:**

There are a few optional clarifications that could improve clarity of the paper, but they are not blocking:
- Clarify AttnCon implementation details in the main text (layers/heads aggregation and normalization), rather than leaving them primarily in the rebuttal.
- Add a brief discussion positioning RSQ relative to AWQ in the Related Work section.
- Explicitly summarize calibration‑time overhead vs. QuaRot in the main paper.

**Audience:**

Yes

**Audience Explanation:**

At least some individuals in TMLR’s audience would be interested in this paper. Specifically:
- Researchers in efficient machine learning and model compression will find the work relevant, as it introduces an importance‑aware refinement to post‑training quantization that yields consistent gains in low‑bit regimes without retraining.
- Practitioners deploying LLMs on constrained hardware are likely to benefit from the proposed framework, which is compatible with existing GPTQ‑style pipelines and preserves inference efficiency while improving accuracy.
- Methodological researchers working on second‑order optimization, quantization objectives, and scaling laws may find the insight (i.e., uniform token reconstruction is suboptimal and attention‑derived importance can guide better approximations) interesting.

Overall, the paper addresses an important problem in modern ML systems, and its contributions align well with the interests of a significant subset of the TMLR readership.

**Claims And Evidence:**

Yes

**Claims Explanation:**

The claims are supported by clear evidence. In particular:

- The core empirical claim (prioritizing important tokens during layer-wise post‑training quantization leads to better performance than uniform reconstruction) is supported by the proposed pilot study and validated across extensive experiments.
- The proposed RSQ method is evaluated on multiple model families (LLaMA3, Mistral, Qwen2.5), multiple bit‑widths (including challenging 2‑bit and 1.58‑bit regimes), and diverse benchmarks, including long‑context tasks, with consistent improvements over strong baselines such as GPTQ and QuaRot.
- Ablation studies compare different token‑importance strategies and justify the final design choice (attention concentration).
- Reviewers’ concerns regarding theoretical justification (importance‑weighted Hessian), computational overhead, hyperparameter sensitivity, and relation to AWQ were addressed in the rebuttal, including additional derivations, runtime/memory measurements, and new large‑model experiments.

Some aspects of the token‑importance choice remain empirically motivated, but this is acknowledged and does not detract from the overall strength of the evidence. Overall, the experimental design, transparency, and breadth of validation convincingly support the paper’s claims.